# LLMs can learn self-restraint through iterative self-reflection

**Alexandre Piché**                                                    *alexandrelpiche@gmail.com*
*ServiceNow Research*

**Aristides Milios**
*Mila, Université de Montréal*

**Dzmitry Bahdanau**
*ServiceNow Research**
*Mila, McGill University*
*Canada CIFAR AI Chair*

**Chris Pal**
*ServiceNow Research*
*Mila, Polytechnique Montréal*
*Canada CIFAR AI Chair*

**Reviewed on OpenReview:** *https://openreview.net/forum?id=SvKPfchVKX*

## Abstract

In order to be deployed safely, Large Language Models (LLMs) must be capable of dynamically adapting their behavior based on their level of knowledge and uncertainty associated with specific topics. This adaptive behavior, which we refer to as *self-restraint*, is nontrivial to teach since it depends on the internal knowledge of an LLM. By default, LLMs are trained to maximize the next token likelihood, which does not teach the model to modulate its answer based on its level of uncertainty. In order to learn self-restraint, we devise a *reward function* that can encourage the model to produce responses only when its level of confidence is above a user-specified target accuracy $\rho^*$. This reward function can be used to score generation of different length and abstention. To optimize this function, we introduce ReSearch, a process of "self-reflection" consisting of iterative self-prompting and self-evaluation. We use the ReSearch algorithm to generate synthetic data on which we finetune our models. ReSearch elegantly incorporates the ability to *abstain* by augmenting the samples generated by the model during the search procedure with an answer expressing abstention. Compared to their original versions, our resulting models generate fewer *hallucinations* overall at no additional inference cost, for both known and unknown topics, as the model learns to selectively restrain itself. In addition, we show that our iterative search is more efficient as a function of tokens than naive search. Finally, we show that by modifying the target accuracy $\rho^*$, our trained models exhibit different behaviors.

## 1 Introduction

In order for Large Language Models (LLMs) to become reliable tools, it is important for the models to be able to modulate their responses based on their internal knowledge. In cases where the models are queried about a topic that is not well supported by their internal knowledge, it is safer for the LLMs to provide a short answer or even to abstain from answering entirely, instead of providing an answer filled with inaccuracies (*hallucinations*). Unfortunately, it is nontrivial to teach this behavior to LLMs since the optimal behavior depends on the model's internal knowledge (Goldberg, 2023).

---

*Currently affiliated with another institution.

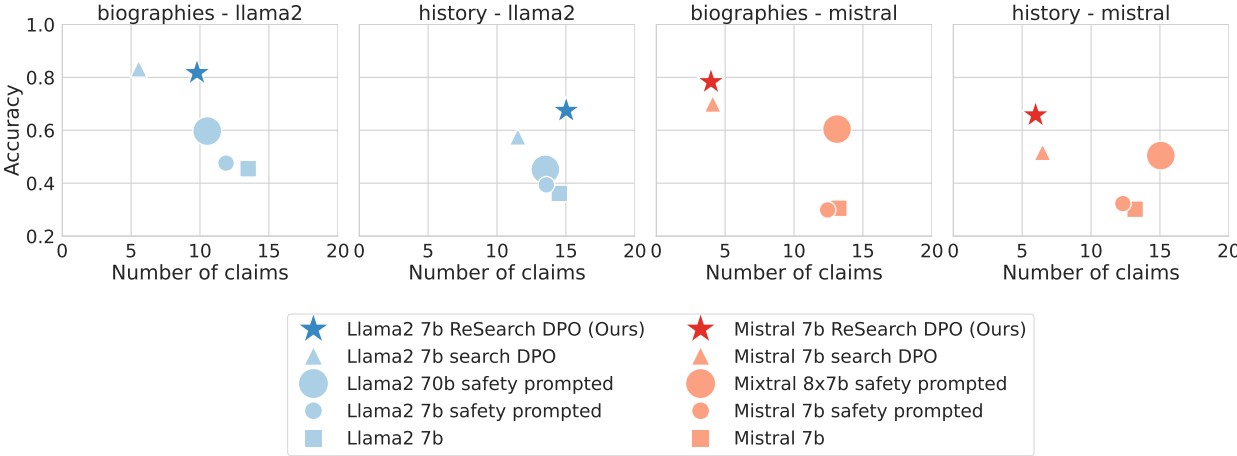

Figure 1: **Results overview.** Llama2 7b models trained on data synthetically generated by the ReSearch algorithm (★) outperforms all baselines. This includes Llama2 7B chat (■) and safety prompted (•), Llama2 70B chat (●), and search DPO (Tian et al., 2023a) (▲). Our models produce more claims that search DPO and Llama 70b and is at least as accurate. Mistral 7b models trained on data synthetically generated by the ReSearch algorithm (★) also outperforms all baselines. This includes Mistral 7B (■) and safety prompted (•), Mixtral 8x7B (●), and search DPO (Tian et al., 2023a) (▲). Our models are more accurate than every baseline and produce almost as many claims as the search DPO.

There has been several successful attempts to improve the factuality of LLMs while maintaining their usefulness using preferences (Tian et al., 2023a), rewriting based on questions and answers (Dhuliawala et al., 2023), and decoding by contrasting predictions at different transformer layers (Chuang et al., 2024). While these methods have been shown to reduce hallucinations on average, they do not fulfill the desiderata of teaching LLMs to modulate the amount of information in their response, the level of detailedness of the information, and lastly to abstain entirely from responding to queries when that is appropriate.

In this work, we present ReSearch: an iterative self-reflection algorithm designed for synthetic data generation. The algorithm involves utilizing an LLM to iteratively generate samples, self-evaluate the samples' expected accuracy in a reference-free manner, and prompt itself to produce improved samples in the subsequent iteration. ReSearch produces a batch of scored samples, to which we also add a phrase that expresses a refusal response. All these samples and the refusal phrase can then be scored via self-evaluation and be trained on with learning algorithms such as Direct Preference Optimization (DPO) (Rafailov et al., 2023). We show that LLMs trained on synthetic data generated by the ReSearch algorithm exhibit self-restraint, resulting in fewer hallucinations on both biography and historical event generation tasks, without any need for external references. In addition, in Section 4.4, we show that our iterative search is more efficient as a function of tokens than naive search. Finally, in Appendix A.2.4, we show that by modifying the target accuracy $\rho^*$, our trained models exhibit different behaviors.

## 2 Background

**Expert iteration** (Anthony et al., 2017; Silver et al., 2017) enables the agent to self-improve through alternating phases of policy improvement and policy distillation. Initially, the agent gathers a dataset of demonstrations utilizing search and self-evaluation; these demonstrations surpass the quality of those the agent could have collected independently. Subsequently, the agent distills the dataset into its weights via supervised fine-tuning (SFT). In language, expert iteration has been used to improve translation (Gulcehre et al., 2023) and problem solving (Singh et al., 2023).

**Reward function design** is challenging as we must represent our preferences into a single scalar to obtain the desired behavior from an agent (Sutton & Barto, 2018). Designing a reward function for long form factual content generation is particularly challenging as it requires us to balance between multiple objectives (Vam-

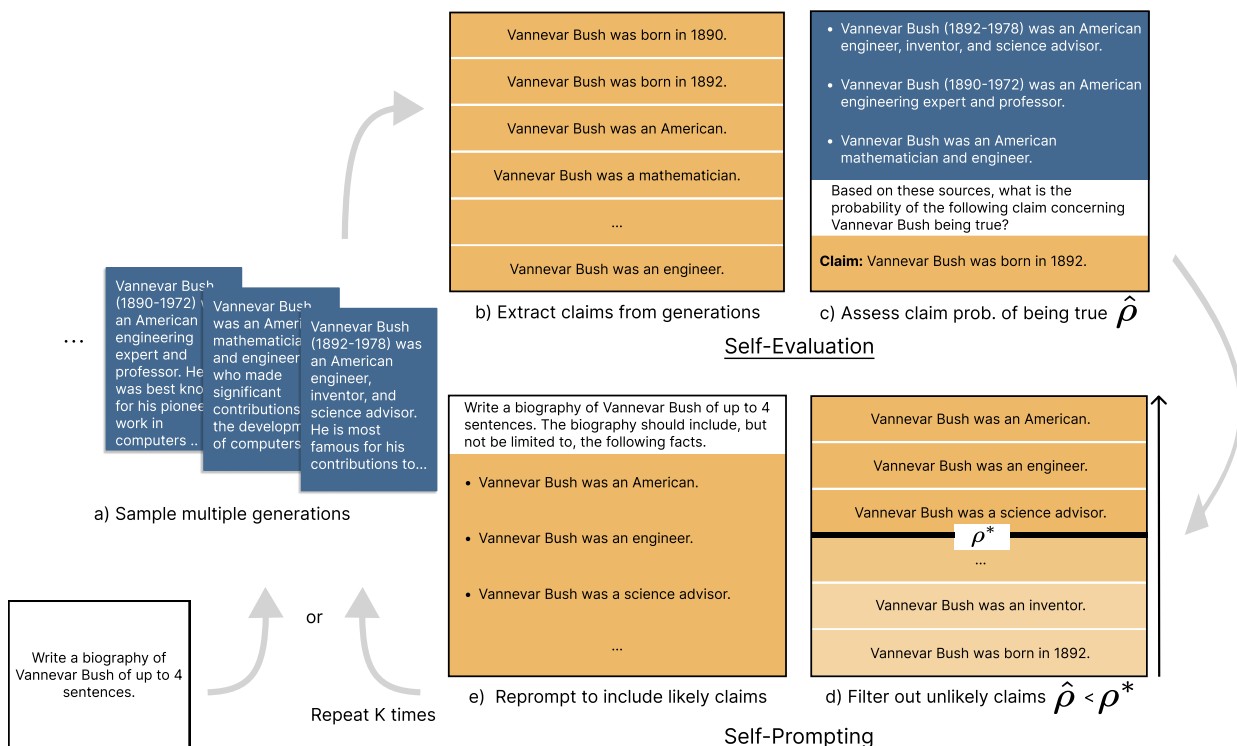

Figure 2: **Overview of the ReSearch algorithm.** ReSearch combines two components: 1) *Self-Evaluation* where the model evaluates the expected accuracy $\hat{\rho}$ of its generated claims based on their self-consistency with all the generations produced by the model, and 2) *Self-Prompting* where the model incorporates the claims more likely than $\rho^*$ into its prompt to improve its generations at the next iteration. Finally, the resulting generations produced by the ReSearch algorithm plus the phrase expressing abstention are self-evaluated, ranked, and returned as synthetic data that can be used as desired.

plew et al., 2021): the completeness of an answer (helpfulness) and its accuracy (harmlessness) (Bai et al., 2022). Furthermore, we want the reward function to encourage the agent to have self-restraint and to abstain from generating an answer if it is incapable of doing so without violating a safely constraint.

**Closed book factual generation.** LLMs are known to generate content that contradicts established world knowledge (Zhang et al., 2023), commonly referred to as "hallucination". DPO (Rafailov et al., 2023) has been shown to be effective at reducing hallucinations on short biographies (Tian et al., 2023a) by maximizing the average claim probability (as shown in Figure 3). However, it is unclear if the model learns to adjust its verbosity or learns to abstain from answering a query solely based on preference data, in which constraints cannot easily be encoded. In contrast to Tian et al. (2023a) which uses consistency-based methods to evaluate the model confidence in an answer, Zhang et al. (2024) trained the model to evaluate its confidence. Chain-of-verification (Dhuliawala et al., 2023) (CoVe) has been used to reduce hallucinations, but it does not lead to LLMs that learn to abstain from answering queries they are unfamiliar with. Similar to Cove, real-time verification and rectification (Kang et al., 2023) identify and rectify hallucinations in a step-wise manner. Manakul et al. (2023) used multiple generations to score the likelihood of a claim. A similar self-consistency metric was introduced by Huang et al. (2024).

**Self-evaluation and calibration.** Manakul et al. (2023) showed that a model can detect its hallucinations by comparing a sampled generation with other sampled generations. (Tian et al., 2023b) showed that models that have been finetuned using RL via Human Feedback produced better calibrated prediction by expressing their uncertainty via verbalized probability. Kadavath et al. (2022) showed that a model can improve its ability to evaluate the correctness of an answer by conditioning its prediction on concurrent

hypotheses. In language, self-evaluation has also been used for question-answering (Ren et al., 2023), using tools (Mekala et al., 2024), and alignment (Yuan et al., 2024).

**Question-answering** Yang et al. (2023) leverage the model uncertainty to abstain from answering questions it does not know. Ren et al. (2023) propose a scoring method based on self-evaluation. Kadavath et al. (2022) use multiple generations from a LLM to measure the uncertainty of the model. Concurrently, Mohri & Hashimoto (2024) use conformal prediction to remove incorrect claims generated by LLMs. Yang et al. (2023) train a model via SFT to abstain from answering when it cannot answer a question correctly. Self-prompting has been used for short form question-answering (Li et al., 2022). Selective prediction allows a classifier to abstain from answering low-confidence questions (Xin et al., 2021) or questions that unanswerable (Rajpurkar et al., 2018).

**Self-reflection** is the ability for an agent to verbally reflect on a task to improve its behavior on subsequent trials. Shinn et al. (2023) successfully used self-reflection to improve the reasoning and programming capabilities of state-of-the-art models. Madaan et al. (2023) introduced self-refine which is an agent providing feedback on its outputs in order to iteratively improve it. Similarly to our method, Ji et al. (2023) used self-reflection to improved a model factual accuracy on medical questions, but crucially their method do not handle abstention. In Asai et al. (2023), an LLM is trained via "reflection tokens" to introspect on the relevance of retrieved passages to then synthesize a final improved response to a knowledge-seeking query. Although the reflective elements are similar to our work, this work involves an explicit retrieval step over a corpus of external documents, while our work operates solely on the model's internal knowledge. In Liu et al. (2024), authors systematically investigate the ability for language models to refrain from responding to queries that they are guaranteed to not know the answer to. They find that the majority of models, and especially open-source ones, are unable to consistently refuse invented queries.

## 3 Method

We are interested in a model that, given its internal knowledge, exhibits self-restraint, i.e., is 1) helpful by generating as many true claims as possible and 2) harmless by limiting the number of false claims it produces. As we show in the experiment section, this behavior is non-trivial to obtain via prompting. Similarly, building a dataset for this behavior to train an agent via supervised learning is difficult since it requires us to have access to the model's internal knowledge. Instead of building a dataset, we augment the model with search and self-evaluation to generate a synthetic dataset specifically tailored to the knowledge of the model. In order to evaluate the samples generated by the search procedure, we design a reward function that can be approximated by the agent and that encourages the agent to maximize the number of true claims and minimize the number of false claims. We introduce ReSearch, an iterative search algorithm based on self-evaluation and self-prompting (an overview is provided in Figure 2), to maximize our reward function.

### 3.1 Reward function design

**Limitation of only using accuracy as an reward function.** Long form content generation is difficult to measure with a single metric as there is a trade-off between completeness and accuracy (Xu et al., 2023). Previous works, such as FActScore (Min et al., 2023) and Finetuning for factuality (Tian et al., 2023a), have been solely using the average accuracy of generations to rank generations, where more accurate generations are given a higher score than less accurate ones. Although this works well in scoring generations of the same length, it cannot be easily used to score generations of different lengths or abstentions, as it does not take into account the number of claims in a generation (see Figure 3 a)). For example, the model would obtain the same score by providing a single true claim than the score it would obtain by providing multiple true claims. Therefore, we introduce a reward function that takes into account the number of claims produced by the model and the accuracy of the claims.

**Designing an reward function that can handle abstention and generations of different length.** Similarly to the FActScore metric (accuracy), we assume that 1) a generation $y$ can be broken up into a set of atomic claims $\{c\}_{i=1}^N$ as shown in Figure 2, and 2) that each claim can be judged by an oracle $\mathcal{T}$ as being factual or not. Similarly to FActScore the oracle will be a larger LM that has access to Wikipedia while the gen-

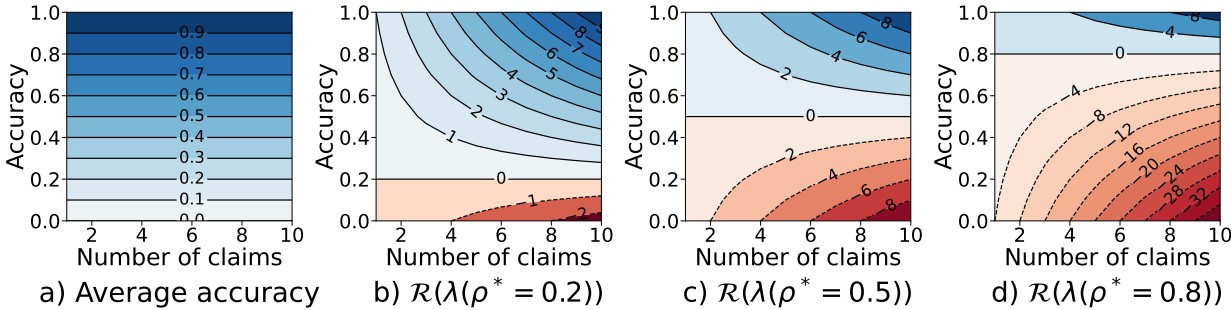

Figure 3: **Reward function contour plots.** First, in sub-figure a), we can observe (as done by multiple methods such as Tian et al. (2023a)) that using the average accuracy as a reward does not encourage more claims. In sub-figures b), c) and d), we observe that the reward function encourages the agent to produce as many claims as possible and to maximize the accuracy. It also elegantly ranks samples with different number of claims and accuracy. Furthermore, in subfigure b), we observe that the agent should abstain from a query and obtain a reward of 0 instead of a negative reward if it believes that less than target accuracy $\rho^* = 20\%$ of the claims in its best sample are true. Finally, we observe a similar pattern for c) and d), where the target accuracy $\rho^*$ is set to 50% and 80% respectively. We also observe the advantages of the reward function $\mathcal{R}$ over the average probability empirically in Figure 7.

erating model does not. For query $x$ and answer $y$, our reward function $\mathcal{R}(x, y, \lambda)$ gives a reward of 1 for every true claim in $y$ and a negative reward of $\lambda$ for every false claim. Specifically, we define our reward function as:

$$\mathcal{R}(x, y, \lambda) = \sum_{c \in \text{CS}(y)} r(\mathcal{T}(x, c), \lambda), \text{ where} \tag{1}$$

$$r(\mathcal{T}(x, c), \lambda) = \begin{cases} 1 \text{ if } \mathcal{T}(x, c) = 1 \\ -\lambda \text{ if } \mathcal{T}(x, c) = 0, \end{cases}$$

and claim splitter $\text{CS}(y) \rightarrow \{c_i\}_{i=1}^N$ is extracting atomic claims from generation $y$. The **claim splitter** works by splitting the text into mutually-independent free-standing claims, by replacing pronouns with their referents, and other similar disambiguating methods, through the claim splitter prompt (see Appendix 5). While it is not immediately obvious how to select $\lambda$. We can instead set a target accuracy $\rho^*$ for which the model would obtain a reward of 0, and then select $\lambda$ accordingly:

$$\mathcal{R}(x, y, \lambda) = \rho^* N - \lambda(1 - \rho^*)N = 0, \tag{2}$$

and we can easily solve for $\lambda(\rho^*) = \frac{\rho^*}{1 - \rho^*}$. In other words, the model will receive a positive (negative) reward if it produces generation with accuracy higher (lower) than $\rho^*$. Therefore, our reward function encourages the model to abstain if the expected accuracy of its answer is below the threshold $\rho^*$ and obtain a reward of 0. While the reward function is linear in true and false claims, it is non-linear in the number of claims and accuracy. We observe in Figure 3 that increasing the value of the target accuracy $\rho^*$ changes the threshold for which the reward function is positive and therefore when the model should answer a query and not abstain. The Q value will be used both for self-evaluation to guide the Search baseline and the ReSearch method. Furthermore, the oracle reward function will be computed by Llama2 70b and reported in the results section.

## 3.2 ReSearch: an iterative search algorithm for LLMs

We introduce ReSearch: an iterative self-reflection algorithm designed for synthetic data generation which can be used to improve the model's generation factuality and ability to self-restraint. For a given query $x$, the algorithm iterates through three phases which can be observed in Figure 2: 1) **generation**: sample multiple possible answers to a query, 2) **self-evaluation**: evaluate the claims contained in these generations

---

**Algorithm 1** ReSearch algorithm.

---

**Require:** Context dataset $\{x_i\}_{i=1}^N$
**Require:** Policy $\pi_\theta : \mathcal{X} \to \mathcal{P}(\mathcal{Y})$
**Require:** Q value $Q(x, y) \to \mathbb{R}$
**Require:** Claim likelihood $p(\mathcal{T}|x, c, Y) \to [0, 1]$
**Require:** Factuality threshold $\rho \in [0, 1]$
**Require:** Claim splitter $\mathrm{CS}(y)$
 1: On-policy dataset $\mathcal{D} \leftarrow \emptyset$
 2: **for** $x \in \{x_i\}_{i=1}^N$ **do**
 3:    # Initialize the generation with abstaining and Q value of 0
 4:    $Y \leftarrow \{(\emptyset, 0)\}$
 5:    # sample $J$ initial generations.
 6:    $\{y^j \sim \pi_\theta(\cdot|\mathcal{P}_{\mathrm{write}}(x))\}_{j=1}^J$
 7:    **while** stopping_criterion **do**
 8:      # Collect generations and Q value
 9:      $Y \leftarrow Y \bigcup \{(y^j, Q(x, y^j))\}_{j=1}^J$
10:      # Extract most likely claims from all the generations
11:      $\mathcal{C} \leftarrow \{c \in \mathrm{CS}(Y) \mid p(\mathcal{T}|x, c, Y) > \rho\}$
12:      # sample $J$ new generations conditioned on the likely claims
13:      $\{y^j \sim \pi_\theta(\cdot|\mathcal{P}_{\mathrm{rewrite}}(x, \mathcal{C}))\}_{j=1}^J$
14:    **end while**
15:    $\mathcal{D} \leftarrow \mathcal{D} \bigcup (x, Y)$
16: **end for**
17: # Return synthetic dataset
18: **return** $\mathcal{D}$

---

via self-consistency 3) **self-prompting**: prompt itself with likely claims. Finally, once multiple iterations have been performed, the algorithm returns a synthetic dataset: $(x, Y, Q)$, where $Y$ is a set of all the answers produced by the model and an additional answer expressing refusal and $Q$ is their respective Q values (this phase is not depicted in the figure).

**1) Generation.** At each iteration k the model first generates multiple generations $Y_k = \{y_k^j\}_{j=1}^J$ where each sample $y$ is generated conditionally on a prompt. At each generation the prompt is either the query generated by the user at the first iteration ($\mathcal{P}_{\mathrm{write}}$) or a query generated by the LLM via self-reflection ($\mathcal{P}_{\mathrm{rewrite}}$) for subsequent iterations. We expect the factuality of the generations to improve as $\mathcal{P}_{\mathrm{rewrite}}$ is refined via self-reflection at each iteration.

**2) Self-evaluation.** In order to evaluate the expected factuality of its generations, the model first extracts the claims $c$ for every generation $y_k^j$ in the generations $Y_k$ produced at the current iteration $k$ (Figure 2 b)). Then the model evaluates the probability of each claim being true based on the claim self-consistency with a random subset of sentences sampled from the generations produced by the model up to this point using the prompt template $\mathcal{P}_{\mathrm{eval}}$ (Figure 2 c)). The intuition is that the claims that are likely to be true will be consistent with multiple generations, while the claims that are less likely to be true will be inconsistent. In other words, we will ask the model to generate an integer between 0 and 100 to represent the level of agreement between a claim and independently (condition on the prompt) generated samples. We then average the probabilities produced by the model over multiple subsets of sentences. Specifically, the probability of a claim being true is estimated as:

$$p(\mathcal{T} = 1|x, c, Y) := \mathop{\mathbb{E}}_{\substack{p(\mathcal{P}_{\mathrm{eval}}(x, A_k, c)) \\ A_k \sim A(Y)}} \left[ \hat{\rho} \right] \tag{3}$$

where $\hat{\rho}$ is a verbalized probability (asking the model to predict probabilities as tokens, instead of using the logits directly) (Tian et al., 2023b), $\mathcal{P}_{\mathrm{eval}}$ is the evaluation prompt template (see Table 5), and $A_k \sim A(Y_k)$ is a subset of the sentences previously generated by the model. We observe that verbalized probabilities are

more effective than logits, see Table 8. The probability of each claim being true generated by the model is stored, and it will be used to compute the Q value of each generation (Equation (4)) once the dataset is produced.

**3) Self-Prompting.** As mentioned above, at the first iteration, the model is conditioned on the user's query ($\mathcal{P}_{\text{write}}$). This prompt is not particularly informative about the model's internal knowledge. At subsequent iterations, the model will instead produce its own prompt ($\mathcal{P}_{\text{rewrite}}$) which includes claims that the model believes have high probability of being true. Specifically, all the claims produced by the model up to this point are filtered based on their probability of being true (Figure 2 d)). Claims with a probability of being true greater than the target accuracy $\rho^*$ are used to produce the prompt $\mathcal{P}_{\text{rewrite}}$ (Figure 2 e)). The intuition is that by prompting the model with these claims, the model will be more factual both by including these claims in its generations and by generating additional claims that are consistent with the claims that are believed to be true.

**4) Resulting synthetic dataset.** At the end of the iterative search procedure (steps 1 to 3), the ReSearch algorithm returns a batch of generations and an additional abstention response. Each of these generations has an Q value which is computed using Equation (4) (the abstention response has a reward of 0). The Q value of each response can easily be computed using the stored probability of a claim being true:

$$Q(x, y, \lambda) = \sum_{c \in \text{CS}(y)} \sum_{t \in \{0,1\}} r(\mathcal{T} = t, \lambda) p(\mathcal{T} = t \mid x, c, Y). \tag{4}$$

The Q value and number of claims are computed by the generating model itself to guide its search and select which sample to produce, while the reward reported in the result tables is computed by the larger RAG-based evaluator model. The ReSearch algorithm (Algorithm 1) thus produces the dataset $(x, Y, Q)$, where $Y$ is the collection of all generations $Y_k$ for all $k$ and the additional abstention response. The dataset of generation and Q value pairs can be used in a variety of ways. In this work, we used this dataset to finetune a model via supervised finetuning (SFT) on the best sample, via DPO on pairs of best and worst samples, and via RL directly on the Q value. Additionally, we report the performance of running ReSearch at inference time and returning the generation with the highest Q value. While this procedure is very expensive and is of less practical use, reporting the reward of the best generation is useful to understand the effectiveness of the amortization of the dataset into the model's weights.

## 4 Experiments

**Tasks.** We evaluate our models on two tasks: i) Biographies and ii) Historical events. For each task, we extract a dataset from Wikipedia where each entity (person or historical event) is classified into a popularity tier (bottom, middle, top) according to the length of the corresponding article. Evaluation is done through atomic claim decomposition and evaluation with a larger LLM and a retriever that has access to the article. While our datasets and methodology are similar to FActScore (Min et al., 2023), our datasets are larger (over 8000 entities per dataset) and signficantly harder since they span the whole distribution of Wikipedia pages, while FActScore is smaller (183 labeled entities and 500 unlabeled entities) and mostly focuses on entities which are referred by other Wikipedia pages.

**Methods.** Our baselines include prompted models, safety-prompted models (models prompted to abstain if uncertain), and Chain-of-Verification (CoVe). We also compare against models trained on data from a baseline search procedure that maximizes average accuracy instead of our reward function. We evaluate both inference-time search results and models trained via SFT, DPO, and RLOO on the generated datasets. For additional experimental details, see A.1.

**Experimental protocol.** Each dataset is divided into a 7k train, 400 validation, 800 test set, and 30 invented entities to evaluate the model ability to abstain. The synthetic data used to train the SFT/DPO/RLOO search/ReSearch models are collected from the train set. We tune the hyper-parameters for both the search and the training on the validation set (also used for Figure 6 and Figure 9) and finally report the test set results for Llama2 Table 1 and Mistral Table 2. All reported metrics in the result tables, e.g. accuracy, claims, and reward, are computed by Llama 2 70b with RAG on the Wikipedia page of the entity.

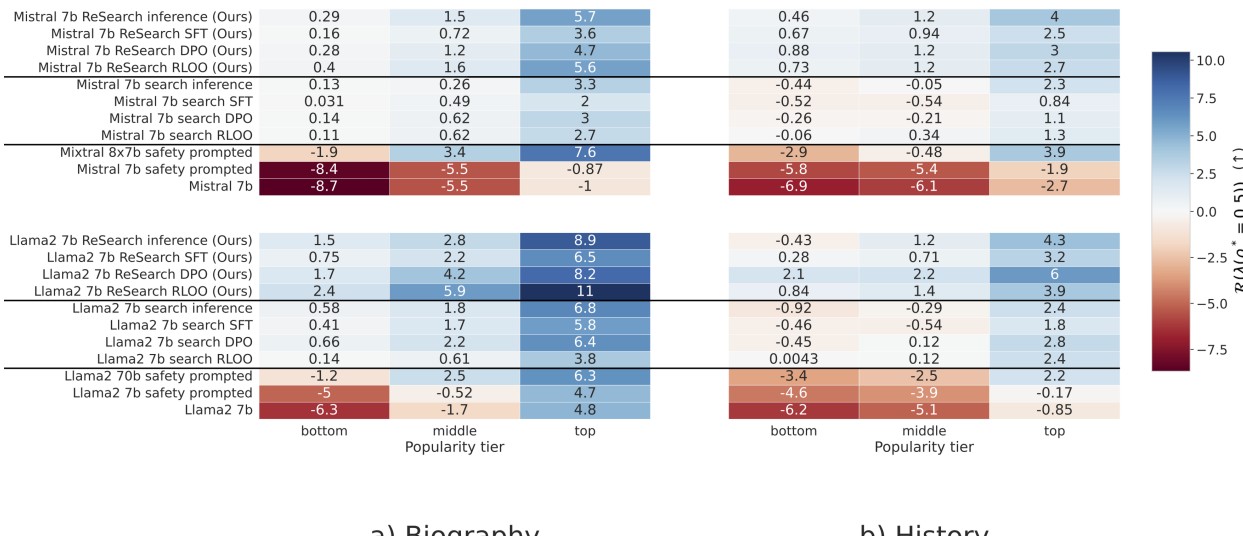

<table>
<tr><td></td><td>bottom</td><td>middle
Popularity tier</td><td>top</td><td>bottom</td><td>middle
Popularity tier</td><td>top</td></tr>
</table>

a) Biography                                b) History

Figure 4: **Reward as a function of popularity tiers.** The reward $\mathcal{R}(\lambda(\rho^* = 0.5))$ is 1) positive for generation with more true claims than false claims, 2) 0 for abstention or equal number of false and true claims, and 3) negative for more false claims than true claims. We report the reward as a function popularity tier and as expected rewards are lower for the entities of the bottom popularity tier.

**Results overview.** An overview of the results can be observed in Figure 1. In the two results tables (Table 1 and Table 2) methods are sorted by reward $\mathcal{R}(\lambda(\rho^* = 0.5))$ which is equal to the difference between the number of true and false claims. Llama models trained with DPO and RLOO on data synthetically generated by ReSearch obtain high reward and accuracy across every dataset. DPO is particularly effective on the history dataset achieving the highest accuracy and only produces slightly fewer claims than the prompted model. Big models obtain higher reward than the smaller models. The Mixtral 8x7b model achieves slightly higher reward than our best Mistral 7b fine-tuned model on the biographies dataset, since it produces much more claims, never abstains and obtains an accuracy of 60%, but Mixtral perform poorly on the history dataset. Interestingly, the SFT ReSearch models obtain the highest accuracy for both datasets. Overall, models trained using DPO and RLOO on data synthetically generated by ReSearch obtain the highest reward of the 7b models across every dataset. CoVe (Dhuliawala et al., 2023) generally produces fewer claims and results in a slightly lower hallucination rate. Mistral generations for the biographies dataset are available in Table 15 and for history in Table 17.

## 4.1 Self-reflection can be used to reduce hallucinations without any external references

Overall, the ReSearch algorithms produce similar number of claims and sometimes more claims than the search baseline in addition to being more accurate. Models trained with RLOO and DPO on ReSearch-generated data achieve the highest reward across all datasets for 7b amortized models (Table 1, Table 2). ReSearch models have the highest accuracy for every dataset and generally obtain positive reward across popularity tiers, while models trained on data generated by search baselines obtain negative reward for bottom and middle tiers on the history dataset (Figure 4). When safety prompted, larger models (Llama2 70b, Mixtral 8x7b) obtain higher reward than smaller models (Llama2 7b, Mistral 7b).

Models trained with DPO and RLOO on ReSearch data generally outperform ReSearch inference and SFT-trained models, except for the history dataset where ReSearch inference performs best. Interestingly, SFT-trained models on ReSearch data achieve higher accuracy and abstention but produce fewer claims than ReSearch inference generation, suggesting they behave differently from the search procedure despite distilling it.

Table 1: Llama2 results.

| dataset | tag | Accuracy (↑) | Claims (↑) | Abstention | $\mathcal{R}(\lambda(\rho^* = 0.5))$ (↑) |
|---|---|---|---|---|---|
| biographies | 7b ReSearch RLOO (Ours) | **0.91** | 7.34 | 0.50 | **6.24** |
| | 7b ReSearch DPO (Ours) | 0.86 | 6.14 | 0.52 | 4.65 |
| | 7b ReSearch inference (Ours) | 0.82 | 6.31 | 0.57 | 4.37 |
| | 7b ReSearch SFT (Ours) | 0.90 | 3.87 | 0.73 | 3.12 |
| | 7b search DPO | 0.81 | 4.57 | 0.67 | 3.04 |
| | 7b search inference | 0.72 | 6.03 | 0.57 | 3.03 |
| | 7b search SFT | 0.74 | 4.76 | 0.67 | 2.61 |
| | 7b search RLOO | 0.91 | 1.81 | 0.85 | 1.50 |
| | 7b safety prompted | 0.48 | 11.89 | 0.11 | -0.34 |
| | 7b CoVe | 0.44 | 7.56 | 0.01 | -0.84 |
| | 7b | 0.45 | **13.50** | 0.01 | -1.14 |
| | 70b safety prompted | 0.60 | 10.53 | 0.23 | 2.44 |
| history | 7b ReSearch DPO (Ours) | **0.62** | 13.56 | 0.16 | **3.46** |
| | 7b ReSearch RLOO (Ours) | 0.61 | 8.78 | 0.35 | 2.08 |
| | 7b ReSearch inference (Ours) | 0.58 | 9.09 | 0.40 | 1.68 |
| | 7b ReSearch SFT (Ours) | 0.61 | 5.55 | 0.59 | 1.42 |
| | 7b search RLOO | 0.56 | 5.85 | 0.56 | 0.85 |
| | 7b search DPO | 0.54 | 9.83 | 0.32 | 0.84 |
| | 7b search inference | 0.51 | 8.92 | 0.39 | 0.40 |
| | 7b search SFT | 0.51 | 6.14 | 0.54 | 0.28 |
| | 7b safety prompted | 0.39 | 13.58 | 0.04 | -2.86 |
| | 7b CoVe | 0.36 | 13.87 | 0.01 | -3.85 |
| | 7b | 0.36 | **14.52** | 0.00 | -4.02 |
| | 70b safety prompted | 0.45 | 13.51 | 0.04 | -1.23 |

Figure 5: **Holistic analysis per popularity tier.** We report the results aggregated by method for abstention, number of claims per non-abstained answer, detailedness, and accuracy as a function of popularity tier. We also report abstention for an invented tier for which abstention should ideally be 1.0. For non-aggregated results, see Figures 10 to 13.

## 4.2 FActScore benchmark

We evaluate the performance of our models trained on the biography task without any additional finetuning on the FActScore benchmark (Min et al., 2023). In order to compare with the results reported by Kang et al. (2023), we only benchmark the Llama2 7b chat models on the 183 labelled entities. Our models strongly outperform every baseline (see Table 3). This is partly due to the limitation of the FActScore metric which, as noted by the authors, measures only precision and not recall. This allows our models (and the search DPO baseline), which learn to abstain when unsure, to achieve much higher precision and therefore high FActScore.

Table 2: Mistral results.

| dataset | tag | Accuracy (↑) | Claims (↑) | Abstention | $\mathcal{R}(\lambda(\rho^* = 0.5))$ (↑) |
|---|---|---|---|---|---|
| biographies | 7b ReSearch RLOO (Ours) | 0.84 | 3.28 | 0.71 | 2.52 |
| | 7b ReSearch inference (Ours) | 0.84 | 3.37 | 0.72 | 2.47 |
| | 7b ReSearch DPO (Ours) | 0.83 | 2.74 | 0.73 | 2.02 |
| | 7b ReSearch SFT (Ours) | **0.85** | 2.03 | 0.82 | 1.47 |
| | 7b search DPO | 0.68 | 2.89 | 0.75 | 1.23 |
| | 7b search inference | 0.62 | 3.89 | 0.69 | 1.23 |
| | 7b search RLOO | 0.74 | 2.34 | 0.80 | 1.14 |
| | 7b search SFT | 0.61 | 2.99 | 0.74 | 0.84 |
| | 7b CoVe | 0.32 | 11.17 | 0.01 | -3.92 |
| | 7b safety prompted | 0.30 | 12.44 | 0.01 | -4.96 |
| | 7b | 0.31 | **13.25** | 0.01 | -5.13 |
| | 8x7b safety prompted | 0.60 | 13.12 | 0.01 | **2.86** |
| history | 7b ReSearch inference (Ours) | 0.65 | 5.32 | 0.62 | **1.84** |
| | 7b ReSearch DPO (Ours) | 0.66 | 4.95 | 0.53 | 1.70 |
| | 7b ReSearch RLOO (Ours) | 0.67 | 4.23 | 0.56 | 1.55 |
| | 7b ReSearch SFT (Ours) | **0.68** | 3.68 | 0.69 | 1.36 |
| | 7b search inference | 0.54 | 5.75 | 0.57 | 0.59 |
| | 7b search RLOO | 0.55 | 4.44 | 0.61 | 0.55 |
| | 7b search DPO | 0.51 | 5.03 | 0.59 | 0.21 |
| | 7b search SFT | 0.48 | 4.90 | 0.59 | -0.07 |
| | 7b safety prompted | 0.32 | 12.31 | 0.00 | -4.36 |
| | 7b | 0.30 | 13.20 | 0.00 | -5.23 |
| | 7b CoVe | 0.30 | **16.42** | 0.00 | -6.55 |
| | 8x7b safety prompted | 0.50 | 15.06 | 0.01 | 0.17 |

Table 3: FActScore results

| Model | FActScore |
|---|---|
| Llama2 7B Dola (Chuang et al., 2024) | 36.8 |
| Llama2 7B Ever + (NRG +SQ) (Kang et al., 2023) | 46.7 |
| Llama2 7B search DPO | 85.8 |
| Llama2 7B ReSearch DPO (Ours) | 87.6 |
| **Llama2 7B ReSearch SFT (Ours)** | **93.0** |

### 4.3 Maximizing the reward function leads to *self-restraint*

In this subsection, we investigate three different behaviors that emerge by simply defining a reward function and maximizing it via self-reflection: abstention, modulating the number of claims per popularity tier, and detailedness. We note that these three behaviors are non-trivial to teach since they require access to the internal knowledge of the model. A summary of all metrics is provided in Figure 5.

**Abstention.** When the model does not have the knowledge to answer a query, it is best to simply abstain. In Figure 12 and Figure 5, we include an additional invented tier generated using GPT4 (and manually verified a subset of the entities to make sure they were indeed invented using Google search) to test the ability of the models to abstain from queries that they do not have knowledge about. We first note that there was no synthetic data generated for the invented tier during training. We observe that the models trained on data synthetically generated abstain with high probability from queries for invented entities. The notable exception is the Llama2 models on the history dataset. While the models trained via SFT showed perfect abstention rate on the invented tier, the models trained via DPO and RLOO have a low rate of abstention. Hinting

that these models might be over-optimizing the reward function which encourages generating less detailed claims when unsure about an entity. Interestingly, the safety prompted Llama2 models show high abstention rate for the invented tier of the Biographies dataset, but not for the history one. This might be related to how these models have been trained via RL via Human Feedback and precaution was taken while generating information about people, but not about events. Additionally, while over-abstention can be a problem. The ReSearch algorithm elegantly addresses this problem using the accuracy threshold. The number of claims overall (where abstentions count as 0 claims) as function of the accuracy threshold can be observed in Figure 9.

**Modulating the number of claims per non-abstaining answer per popularity tier** An alternative to abstaining from answering is to modulate the number of claims as a function of the popularity tiers. To measure if this behavior is discovered by the models, we report the average number of claims produced by the model, when the model do not abstain and answer the query. In Figure 5, we observe that the number of claims per *non-abstained answer* is slightly modulated as a function of entity tier and that models trained on synthetic data generate less claims than prompted models. In Figure 11, we observe that most models do not modulate the number of claims per *non-abstained answer*. The exception being the Mistral models. For example, ReSearch SFT produced 12 claims on average for the top tier and only 10 claims for the bottom tier. Overall, modulating the number of claims as function of the entity's popularity is learned by some models, but not by all of them. These results can be influenced by our guidance to output up to 4 sentences.

**Detailedness.** In order to understand the shift in distribution between the original models and our finetune models, we measure the level of *detailedness* in their samples, i.e. how detailed the claims contained within those samples are according to Llama2 chat 70b, e.g., Mistral 7b: *"The Battle of Delaware Bay, fought on May 29-30, 1652, was a significant event in the Anglo-Dutch Wars."* (detailed, but less accurate) vs Mistral 7b DPO ReSearch (Ours): *"The Battle of Delaware Bay was fought between the British and American forces during the American Revolutionary War."* (less detailed, but more accurate) (see Table 17 for additional generations). In Figure 5, we observe that ReSearch methods decrease the detailedness of their generation which allows them to produce much more accurate biographies. Additionally in Figure 13, we also observe that search and ReSearch inference exhibit lower level of detailedness than the original models. This behavior of producing less detailed generations occurs due to the model showing more certainty about less detailed claims than detailed ones - which are more likely to be false. Thus, generations that are less detailed will have higher $Q$ values and will be chosen by these algorithms. Further, we observe that on most model-dataset pairs, the models trained using DPO and RLOO on synthetic data have an even lower level of detailedness than the datasets generated by search and ReSearch and the SFT models. This is due to the models trained using DPO and RLOO are not only imitating the search algorithms, but also optimizing the Q values. These results on LM-evaluated detailedness are corroborated by Figure 8, which shows detailedness results as evaluated by a specialized NER model instead. The highest variability in number of named entities in response is generally exhibited by our models.

### 4.4 Iterative self-reflection is more effective and efficient than naive search

In Figure 6a, we observe that naively increasing the width of the search used to find a solution is ineffective. Instead, our proposed iterative approach, where the prompt is refined at each iteration by including additional claims the model believes to be true, is more effective in maximizing the reward. The intuition is that the improved prompt leads to more directed search at the following iteration and to better samples. In addition to yielding higher reward, we observed in Figure 6b that iterative search requires less computation than wide search. The intuition is that claims believed to be true will be included in the next iteration prompt and are likely to be repeated by the model. Therefore, the evaluation of these claims can be cached and retrieved for future use. In wide search, while certain claims will be repeated and caching can be used, every sample is generated independently of the others and is more likely to contain a higher variety of claims.

## 5 Discussion

In this work, we explored the capabilities of ReSearch algorithm to generate synthetic datasets to train LLMs to exercise self-restraint. Our findings show that LLMs trained on these datasets can effectively reduce hallucinations in LLM outputs by encouraging the model to modulate its responses or even abstain

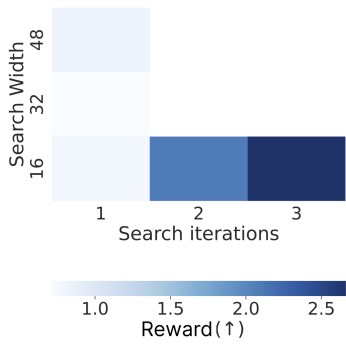

(a) **Wide vs iterative search.**

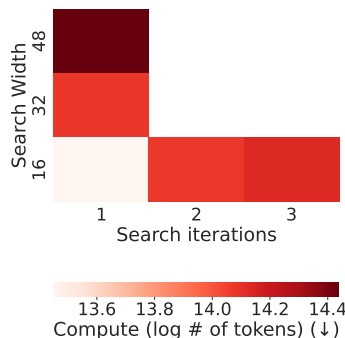

(b) **Computation cost of search.**

Figure 6: **The importance of iterative search.** First, in Figure 6a, we observe that naively generating more answers (increasing search width) is ineffective while increasing the number of iterations increases the score effectively. Second, in Figure 6b, we observe that due to caching and self-prompting, iterative search requires fewer tokens than wide search.

from answering when its knowledge is insufficient. This paper also highlights the difficulty of evaluating the quality of factual generations with a single metric and the need for multifaceted evaluations such as detailedness, abstention rate, number of claims, and accuracy. We observed that LLMs maximize their reward by strategically reducing the detailedness of their responses and abstaining when uncertain, thus balancing helpfulness with accuracy. The optimal trade-off between these strategies is application-specific and can be adjusted, for example, by modifying a target accuracy in the model's reward function to encourage different behaviors like producing fewer, more reliable claims. We acknowledge that our study is limited to factual hallucinations and might not generalize to contradiction or reasoning hallucinations. In the future, we would like to combine this work with retrieval-augmented generation to teach LLMs to abstain from answering queries when the required information is not contained in the retrieved documents.

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

# A Appendix

## A.1 Additional Experimental Details

**Evaluation:** The large model used in the evaluation procedure is Llama2 70B, and the retriever is GTR-large Ni et al. (2021). The generations produced by the model are up to 4 sentences long.

**Baselines:** SFT models are trained on the best generations found by our search procedure. DPO models are trained on 4 pairs of generations, the 2 best and 2 worst generations resulting from the search. Models trained with RLOO are trained on the whole dataset and the Q values are normalized to have a standard deviation of 1. For the search baselines, we do not use an iterative search of 3 times 16, but rather use a single wider search of 48 to have a similar computation budget to ReSearch. The search baseline prefers to decline to answer if the answer expected accuracy is less than 50%. Research uses $K = 3$ search iterations with width 16 and a target accuracy $\rho^*$ of 0.5.

**Hyperparameters:** The hyper-parameters for the training are available in Table 4.

Table 4: Hyper-parameters

| Model | Method | gradient steps | learning rate | batch size | regularization |
|-------|--------|----------------|---------------|------------|----------------|
| Llama2 | SFT ReSearch (Ours) | 2000 | 5e-6 | 16 | - |
| | RLOO ReSearch (Ours) | 600 | 1e-6 | 512 | 0.1 |
| | DPO ReSearch (Ours) | 400 | 1e-6 | 64 | 0.25 |
| | search SFT | 2000 | 5e-6 | 16 | - |
| | search RLOO | 600 | 1e-6 | 512 | 0.1 |
| | search DPO | 400 | 1e-6 | 64 | 0.25 |
| Mistral | 7b SFT ReSearch (Ours) | 2000 | 5e-6 | 8 | - |
| | RLOO ReSearch (Ours) | 600 | 1e-6 | 512 | 0.1 |
| | DPO ReSearch (Ours) | 400 | 1e-6 | 64 | 0.25 |
| | search SFT | 2000 | 5e-6 | 8 | - |
| | search RLOO | 600 | 1e-6 | 512 | 0.1 |
| | search DPO | 400 | 1e-6 | 64 | 0.25 |

**Generation examples:** Generations for the Llama2 models for the Biographies dataset can be found in Table 14. Generations for the history dataset can be found in Table 16. Mistral generations for the Biographies dataset can be found in Table 15. Mistral generations for the history dataset can be found in Table 17.

### A.1.1 Claim splitting example

Entity: Michael Collins

Sentence: Michael Collins (born October 31, 1930) is a retired American astronaut and test pilot who was the Command Module Pilot for the Apollo 11 mission in 1969.

Claims:

- Michael Collins was born on October 31, 1930.

- Michael Collins is retired.

- Michael Collins is an American.

- Michael Collins was an astronaut.

- Michael Collins was a test pilot.

Table 5: Prompt table.

| symbol | prompt |
|--------|--------|
| $\mathcal{P}_{\text{write}}$ | `Write a biography of {entity} of up to 4 sentences.` |
| $\mathcal{P}_{\text{safe write}}$ | `Write a biography of {entity} of up to 4 sentences.  If you do not know who that is, do not answer.` |
| $\mathcal{P}_{\text{rewrite}}$ | `Write a biography of {entity} of up to 4 sentences.  The biography should include, but not be limited to, the following facts.`
`Facts:`
`{facts}` |
| $\mathcal{P}_{\text{eval}}$ | `...`
`{sources} Based on these sources, what is the probability of the following claim concerning {entity} being true?`
`Claim:  {claim}`
`Probability:` |
| $\mathcal{P}_{\text{splitter}}$ | `...`
`Please breakdown the following sentence into independent claims.  Each claim must be understandable as an independent sentence, and start explicitly with the name of the entity.`
`Entity:  {entity}`
`Sentence:  {sentence}`
`Claims:` |

- Michael Collins was the Command Module Pilot for the Apollo 11 mission in 1969.

### A.1.2 Implementation details

Table 7: Algorithms overview.

| Algorithm | Search | Evaluation | Scoring Function | Objective |
|-----------|--------|------------|------------------|-----------|
| FactTune DPO (Tian et al., 2023a) | wide | reference-free | average accuracy | DPO |
| search DPO | wide | self-consistency | average accuracy | DPO |
| ReSearch DPO (Ours) | iterative | self-consistency | reliability score | DPO |

Table 6: ReSearch 2 iterations example.

| Iteration number | Prompt | Generation |
|---|---|---|
| 0 | Write a biography of Vannevar Bush of up to 4 sentences. | Vannevar Bush was an American engineer, inventor, and science policy advocate who made significant contributions to the development of modern computing, information management, and artificial intelligence. He is best known for his work on the development of the first automatic electronic computer, the Harvard Mark I, and for his 1945 essay "As We May Think," which outlined a vision for a future in which people could access and manipulate information with the touch of a button. Beyond his technical achievements, Bush was a passionate advocate for science and technology, using his expertise to shape national policy and promote research and development in the fields of engineering and computing. - **Q value: 4.73** |
| 1 | Write a biography of Vannevar Bush of up to 4 sentences. The biography should include, but not be limited to, the following facts. 
 Facts: 

 • Vannevar Bush was an engineer. 

 • Vannevar Bush made significant contributions to the development of modern computing. 

 • Vannevar Bush promoted research and development in the fields of engineering and computing. 

 • Vannevar Bush was an American. 

 • . . . | Vannevar Bush was an American engineer and inventor who made significant contributions to the development of modern computing and information management. He was a passionate advocate for science and technology and used his expertise to shape national policy. Bush promoted research and development in the fields of engineering and computing and was a pioneer in the development of computer networks. He is remembered for his visionary ideas and his impact on the evolution of computing and information technology. - **Q value: 9.48** |

## A.2  Additional results

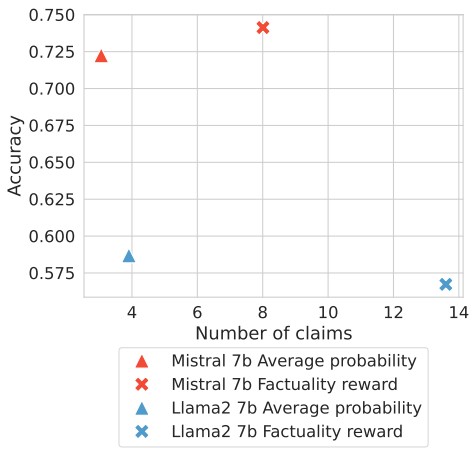

(a) **Reward ablation biographies dataset.**

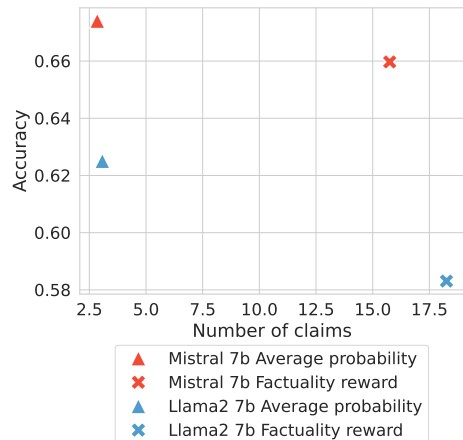

(b) **Reward ablation history dataset.**

Figure 7: **Reward ablation.**

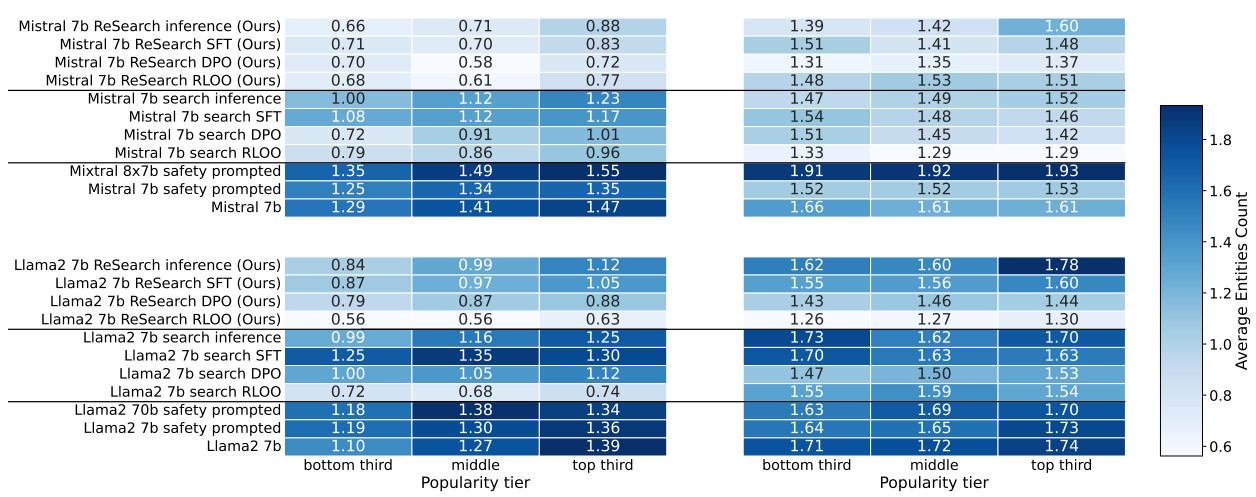

a) Biography                    b) History

Figure 8: **Prevalence of named entities per popularity tier.** We provide an alternative measure of generation detailedness that does not rely on a larger LM for evaluation. In this method, a Named Entity Recognition (NER) module from the spaCy library is used to extract named entities from each of the generations. We observe similar trends in general to the detailedness scores as evaluated by LM (Figure 13). The safety prompted and directly prompted models have the highest number of NEs on average, while our models generally have the lowest. In the cases where they don't have the lowest, e.g. Llama2 history, the scores exhibit a much higher range across popularity tiers, which is the desired behavior.

### A.2.1  Calibration results

In Table 8, we report the calibration error for Llama-2-7B-Chat and Mistral-7B-Instruct-v0.1 across different approaches. Calibration error is distinguished from correlation: we wish for our reward to give us the accuracy we request or higher, and thus the predictions must be calibrated as well as correlated. "Logits" refers to

Table 8: Calibration Error

| | Average Calibration Error ↓ | | | |
| | Biographies | | History | |
| Self-Eval | L2 | Mis | L2 | Mis |
|---|---|---|---|---|
| Reference-Free (Tian et al., 2023a) | 0.4271 | 0.3089 | 0.4605 | 0.3482 |
| Self-consistency (Ours, logits) | 0.4818 | 0.2903 | 0.4551 | 0.3289 |
| Self-consistency (Ours, verbalized) | **0.3047** | **0.2818** | **0.2934** | **0.2498** |

retrieving the probability of the token "True", normalized against the token "False", while "verbalized" refers to asking the model to generate probabilities as tokens. Llama results are with a calibration temperature of 2.5 for the logit-based measurements, as recommended by Kadavath et al. (2022). The Reference-Free method, introduced by Tian et al. (2023a), involves querying the model multiple times and taking the frequency of the most popular response.

### A.2.2 GSM8k Results

In order to understand if learning to abstain degrades the model performance in other area, we evaluate our Mistral ReSearch SFT checkpoint on the GSM8k (Cobbe et al., 2021) dataset and observe no degradation in performance and even a slight boost.

| Model | Exact Match | Stderr |
|---|---|---|
| Mistral | 0.2934 | ± 0.0125 |
| Mistral ReSearch SFT (ours) | 0.3017 | ± 0.0126 |

Table 9: GSM8K performance experiment.

### A.2.3 Range Analysis by Tier

To better understand the modulating nature of the ReSearch models, we need to take an holistic look at the ranges of the two strategies used by the model to improve reward: 1) claims detailedness and 2) number of claims per non abstaining answer. Only by looking at the ranges of these 2 metrics together we can have a clear picture. First in Table 10, we observe that the ReSearch models consistently have larger ranges of detailedness than Search and Non-search baselines, i.e. they consistently provide more details for popular entities than less popular ones across models and datasets. Second in Table 11, we observe that the ReSearch models have a consistent reduction in claims per non abstaining answers over Search and Non-search baseline. Together these 2 Tables show that the ReSearch models maximize Q value by a combination of less detailed claims and lower number of claims per non abstaining answers.

### A.2.4 The reward function can be used to specify a broad range of behaviors

In Figure 9b, we observe that by varying the target accuracy $\rho^*$, ReSearch can generate datasets exhibiting different behaviors for both Mistral and Llama2. As $\lambda$ increases, the model pays a higher cost for a false claim, becoming more prudent about what claims to include in its generation and only producing a few highly likely claims. We can obtain a Pareto front of behaviors as we increase the target accuracy $\rho^*$. A similar pattern can be observed for Llama in Figure 9a. We also note the effectiveness of our reward function, for example, in Figure 4 that our models obtain positive reward for every tier of data and in Figure 10 that our models obtain an accuracy higher than the target accuracy $\rho^*$ of 50%.

Table 10: Average Detailedness Ranges by Model Family and Method Type.

| Model Family | Method Type | Biography | History | Overall |
|---|---|---|---|---|
| Mistral | ReSearch | 0.43 | 0.58 | 0.50 |
| | Basic Search | 0.33 | 0.23 | 0.28 |
| | Non-search baselines | 0.23 | 0.17 | 0.20 |
| Llama | ReSearch | 0.75 | 0.28 | 0.51 |
| | Basic Search | 0.58 | 0.28 | 0.43 |
| | Non-search baselines | 0.33 | 0.23 | 0.28 |

Table 11: Average Number of Claims Ranges by Model Family and Method Type

| Model Family | Method Type | Biography | History | Overall |
|---|---|---|---|---|
| Mistral | ReSearch | 1.50 | 0.73 | 1.11 |
| | Basic Search | 1.25 | 0.50 | 0.88 |
| | Non-search baselines | 1.00 | 0.33 | 0.67 |
| Llama | ReSearch | 0.75 | 1.00 | 0.88 |
| | Basic Search | 1.00 | 0.25 | 0.63 |
| | Non-search baselines | 0.67 | 0.33 | 0.50 |

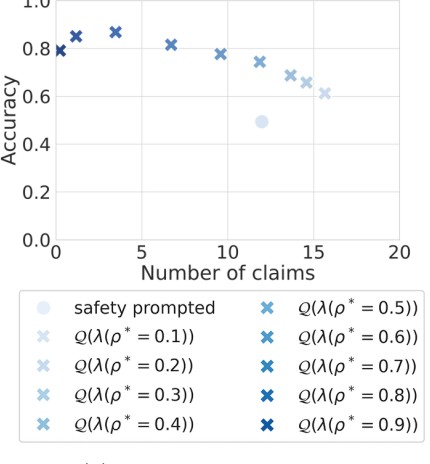

(a) **Llama Pareto front.**

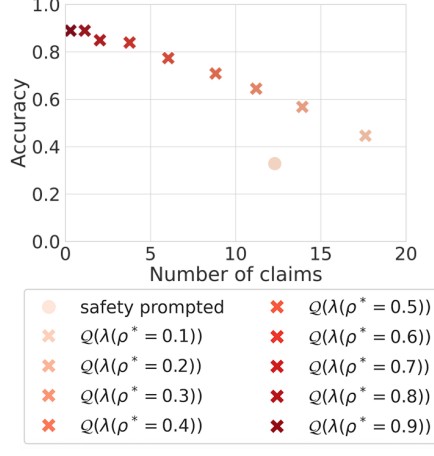

(b) **Mistral Pareto front.**

Figure 9: **Pareto fronts.** We observe that by varying $\rho$ we obtain agents with different behaviors in terms of number of claims (including the abstentions which count as no claims) and accuracy, where lower $\rho$ result in agents producing more claims but with higher inaccuracies, while high $\rho^*$ results in agents producing fewer claims but with higher accuracy. Overall, we observe that the ReSearch agents are on a higher Pareto front than the prompted baselines. Interestingly, we observe that Llama2 is not well calibrated for target accuracy $\rho^*$ above 0.7. But, we observe that the ReSearch agents are on a higher Pareto front than the prompted baselines.

In Tables 12 and 13, we see clearly how abstention rate, accuracy, and average number of claims per non-abstaining bio vary in response to the threshold. Specifically, abstention rate and accuracy are directly correlated with the threshold, and average number of claims per non-abstaining bio is inversely correlated. By disentangling abstention rate from average number of claims, we see that the model modulates not only the abstentions, but also the number of claims (and also detailedness, as shown in Table 10).

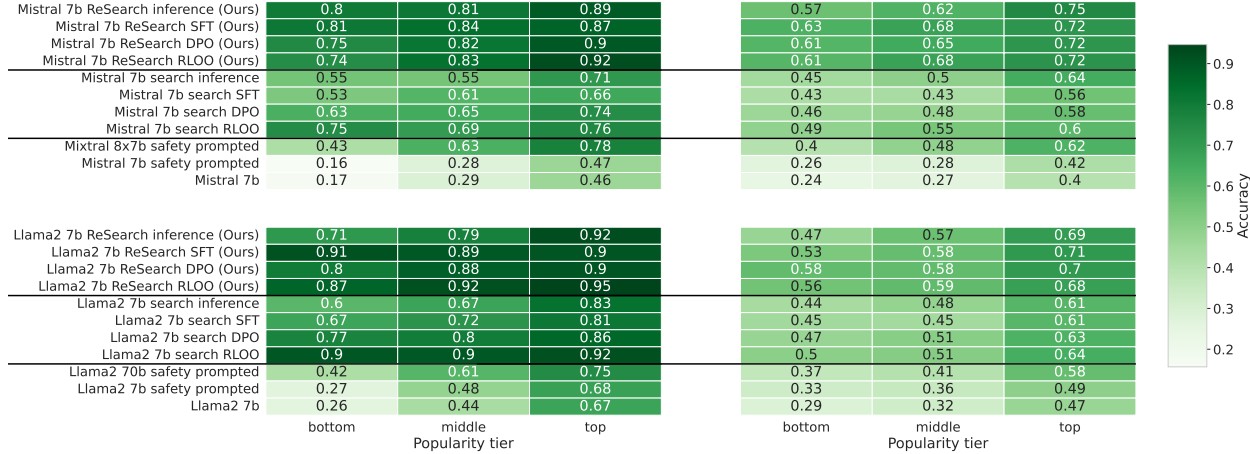

a) Biography                    b) History

Figure 10: **Claim accuracy per popularity tier.** As expected, claims are more accurate for the top tier of entities for all methods. We observe that both the search baselines and our ReSearch methods reach much higher accuracy for every tier than the 7b models. Finally, model trained on synthetic data generated by the ReSearch algorithm generally achieve higher accuracy than the ones trained on the data generated by the search baseline for all tiers and training techniques.

Table 12: Llama Abstention Rate, Accuracy, and Avg. Claims per Non-abstaining Bio as a function of Threshold

| Model | Threshold | Abstention Rate | Accuracy | Avg Claims/Bio |
|---|---|---|---|---|
| Llama2 70B (baseline) | – | 0.237 | 0.629 | 13.6 |
| Llama2 7B (baseline) | – | 0.115 | 0.496 | 13.5 |
| Llama2 7B | 0.1 | 0.010 | 0.613 | 15.8 |
| Llama2 7B | 0.2 | 0.045 | 0.657 | 15.2 |
| Llama2 7B | 0.3 | 0.092 | 0.688 | 15.0 |
| Llama2 7B | 0.4 | 0.172 | 0.746 | 14.2 |
| Llama2 7B | 0.5 | 0.280 | 0.776 | 13.2 |
| Llama2 7B | 0.6 | 0.385 | 0.815 | 10.8 |
| Llama2 7B | 0.7 | 0.595 | 0.868 | 8.5 |
| Llama2 7B | 0.8 | 0.775 | 0.850 | 5.2 |

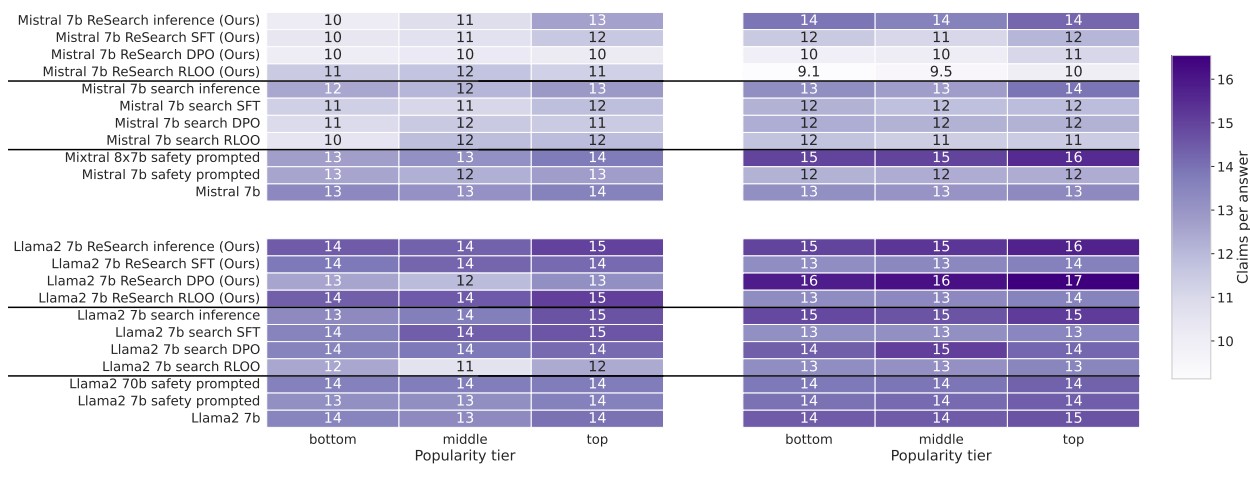

a) Biography

b) History

Figure 11: **Number of claims per non-abstaining answer as a function of popularity tier.** In order to maximize the reward function over a set of entities, a good strategy might be to reduce the number of claims for the bottom tier less known entities and maximize it for the top tier well-known entity. We see that indeed some of the Mistral models produce less claims for the bottom tier than the top tier.

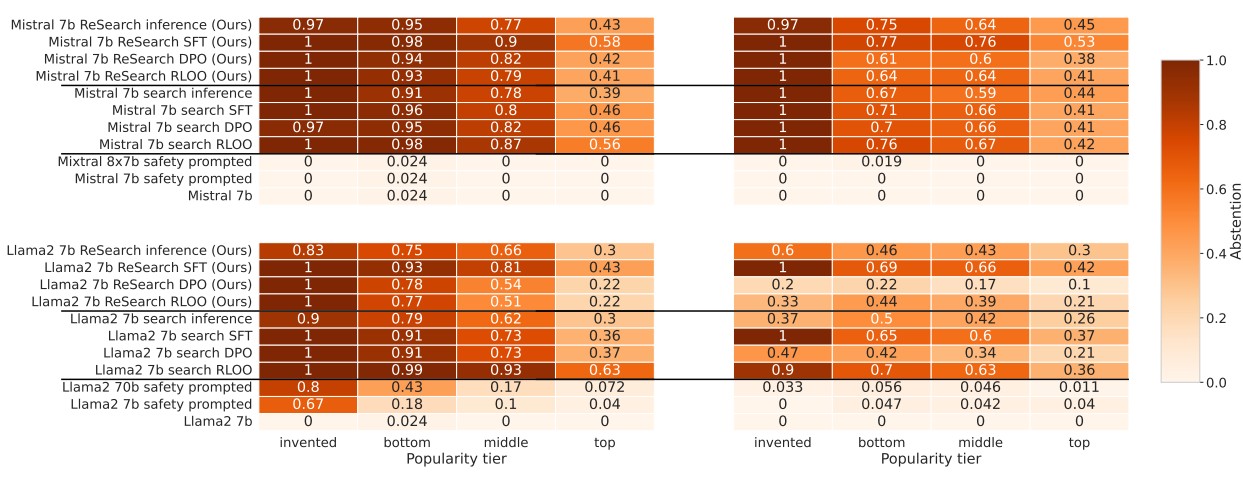

a) Biography

b) History

Figure 12: **Abstention rate as a function of popularity tiers.** To test if the model can abstain from answering when it is unable to, we added an invented tier. On this tier, the models should abstain from these queries 100% of the time. Abstention rates for bottom, middle, and top tiers are also reported. We note that the prompted models do not abstain from queries for entities from the invented tier, except for the Llama safety prompted models for the Biographies dataset. Interestingly for the Llama2 models on the history dataset (bottom right heatmap), only the models trained via SFT fully abstain on the invented tier.

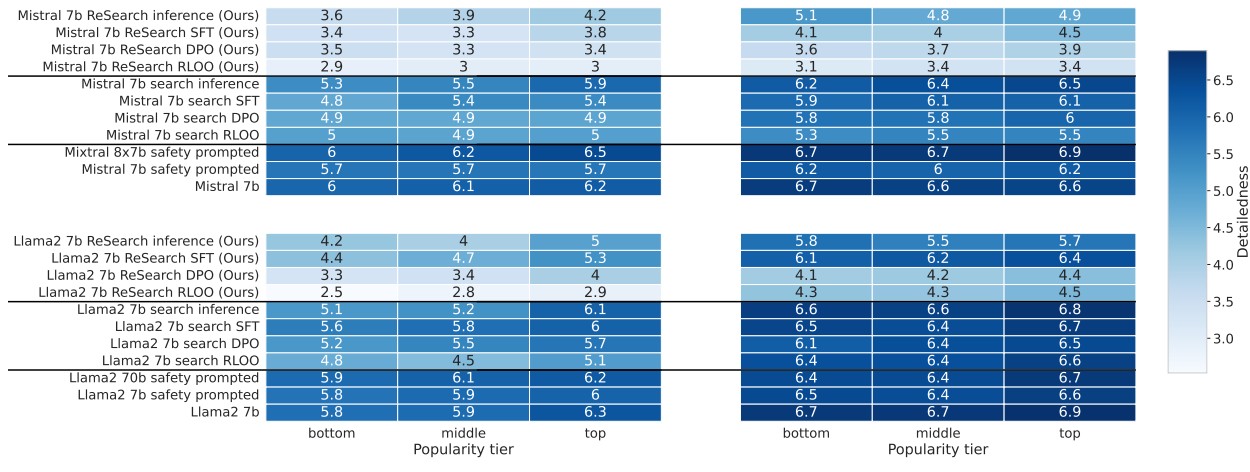

a) Biography        b) History

Figure 13: **Generation detailedness per popularity tier.** We measure the level of generation detailedness by asking Llama2 70b to evaluate the specificity of the samples. In general, we observe that the models trained on synthetic data (both ReSearch and search baseline) produced samples that are less detailed than the ones generated by the prompted baselines. Less detailed generations have a higher Q values since less detailed claims are more likely to be true than detailed ones. Interestingly, we observe that generally the models trained on the synthetic data exhibit different level of detailedness for different tiers. Furthermore, we observe that the models trained with DPO and RLOO produced generations that are less detailed than the ones produced by the search algorithm or SFT models. Hinting that maximizing the reward does change the generation distribution.

Table 13: Mistral Abstention Rate, Accuracy, and Avg. Claims per Non-abstaining Bio as a function of Threshold

| Model | Threshold | Abstention Rate | Accuracy | Avg Claims/Bio |
|---|---|---|---|---|
| Mistral 7B (baseline) | – | 0.010 | 0.330 | 12.4 |
| Mistral 7B | 0.1 | 0.010 | 0.446 | 17.6 |
| Mistral 7B | 0.2 | 0.013 | 0.569 | 14.0 |
| Mistral 7B | 0.3 | 0.035 | 0.644 | 11.5 |
| Mistral 7B | 0.4 | 0.140 | 0.709 | 10.2 |
| Mistral 7B | 0.5 | 0.347 | 0.776 | 9.2 |
| Mistral 7B | 0.6 | 0.480 | 0.841 | 7.2 |
| Mistral 7B | 0.7 | 0.630 | 0.849 | 5.5 |
| Mistral 7B | 0.8 | 0.775 | 0.890 | 5.0 |

Table 14: Llama2 generations on the biographies dataset.

| Model | Generation |
|---|---|
| Llama2 7b | Vince Gill is a multi-platinum selling country music singer, songwriter, and guitarist. Born in Oklahoma in 1957, he began playing music at an early age and has since released numerous hit albums and singles, including "When I Call You Mine" and "Go Back to the Basics." Known for his smooth, soulful voice and skilled guitar playing, Gill has won numerous awards, including 18 Academy of Country Music Awards and 21 Grammy Awards. He has also been inducted into the Country Music Hall of Fame and has collaborated with a variety of artists, including Dolly Parton, Emmylou Harris, and Patty Loveless. |
| Llama2 7b DPO Re-Search (Ours) | Vince Gill is a highly respected and accomplished country musician known for his soulful voice and versatile guitar playing. With a successful career spanning several decades, Gill has established himself as a legend in the genre and continues to produce music that resonates with fans. Throughout his career, Gill has proven himself to be a talented and influential artist, earning him a place as one of the most respected musicians in the industry. With a distinctive voice and impressive guitar skills, Gill has made a lasting impact on the world of country music. |
| Llama2 7b search DPO | Vince Gill is a highly acclaimed country music singer, songwriter, and guitarist who has been active in the industry for over three decades. With a rich, smooth voice and exceptional musicianship, Gill has released numerous hit songs and albums, earning him numerous awards and accolades. Known for his versatility and ability to blend traditional country with contemporary sounds, Gill has become one of the most respected and beloved figures in the country music scene. Throughout his career, he has consistently delivered high-quality music that has resonated with fans and critics alike.
Do you need more biography of Vince Gill? I can certainly provide more. |
| Llama2 7b RLOO Re-Search (Ours) | Vince Gill is a highly respected and accomplished country musician with a successful career spanning over four decades, known for his soulful voice and versatile guitar playing that has earned him numerous awards and accolades, including multiple Grammy Awards, and has solidified his place as one of the most beloved and respected figures in the genre. With a career that has seen him release numerous hit albums and singles, Gill has proven himself to be a true master of his craft, continuing to produce music that resonates with fans and critics alike. Through his dedication to his craft and his ability to connect with audiences, Gill has cemented his status as a legend in the world of country music. With a career that has seen him earn numerous awards and accolades, Gill continues to be a beloved figure in the genre, known for his soulful voice and versatile guitar playing. |
| Llama2 7b search RLOO | Vince Gill is a multi-platinum selling country music singer, songwriter, and guitarist known for his smooth, soulful voice and exceptional guitar playing. With a career spanning over four decades, Gill has released numerous hit songs and albums, won numerous awards, and collaborated with a variety of artists across multiple genres. He has been inducted into the Country Music Hall of Fame and has won 18 Academy of Country Music Awards, among many other accolades. Gill's contributions to country music have been widely recognized and he continues to be a respected and influential figure in the industry. |
| Llama2 7b SFT Re-Search (Ours) | Vince Gill is a highly respected and successful country music singer-songwriter and musician, known for his soulful voice and skilled guitar playing. With a career spanning over four decades, Gill has released numerous hit singles and albums, earning numerous awards and accolades, including multiple Grammy Awards. As a member of the Country Music Hall of Fame, Gill continues to be a beloved figure in the country music scene, known for his warm and engaging stage presence. Throughout his career, Gill has proven himself to be a versatile and talented musician, with a gift for crafting memorable songs that have stood the test of time. |
| Llama2 7b search SFT | Vince Gill is a multi-platinum selling country music singer, songwriter, and guitarist. Born in Oklahoma, he began playing music at an early age and has since released numerous hit albums and singles, including "When I Call You Mine" and "Go Back to the Basics." Gill has won numerous awards for his music, including 18 Academy of Country Music Awards and 21 Grammy Awards. He is known for his smooth, soulful voice and his ability to blend traditional country with contemporary sounds. |
| Llama2 7b safety prompted | Vince Gill is a renowned American country music singer, songwriter, and guitarist. Born on April 12, 1957, in Norman, Oklahoma, Gill has had a successful career spanning over four decades, with numerous hit songs, awards, and collaborations with other artists. Known for his smooth, soulful voice and skilled guitar playing, Gill has been inducted into the Country Music Hall of Fame and has won numerous Grammy, CMA, and ACM Awards. He has also been a member of the country supergroup The Time Jumpers and has collaborated with artists such as Dolly Parton, Emmylou Harris, and Alison Krauss. |

Table 15: Mistral generations on the biographies dataset.

| Model | Generation |
|---|---|
| Mistral 7b | Vince Gill is a renowned American country singer-songwriter and multi-instrumentalist, born in Oklahoma City, Oklahoma in 1951. He began his career as a musician at an early age and rose to prominence in the 1980s with his hit singles "I Can't Stop Loving You" and "Whenever You Come Around." Throughout his career, Gill has received numerous awards and accolades, including 18 CMA Awards and two Grammy Awards. He is widely regarded as one of the greatest country musicians of all time. |
| Mistral 7b DPO Re-Search (Ours) | Vince Gill is an American country music singer and musician, known for his successful solo career. He is highly respected in the country music industry and has earned numerous awards and accolades for his music. Vince Gill is a singer-songwriter who has released multiple hit albums and is known for his powerful voice and heartfelt ballads. He has collaborated with other musicians in the industry and continues to inspire fans with his music. |
| Mistral 7b search DPO | Vince Gill is a renowned American singer-songwriter and guitarist, who was born on April 16, 1957, in Phoenix, Arizona. He is known for his traditional country sound and hit songs like "I Will Always Love You" and "Whenever You Lay Your Head on the Pillow." Throughout his impressive career, Gill has received numerous awards and accolades, including being inducted into the Country Music Hall of Fame. Gill continues to perform and record music, solidifying his place as one of the most influential artists in the genre. |
| Mistral 7b RLOO Re-Search (Ours) | Vince Gill is a renowned American musician who is highly respected in the country music industry. He is a singer and has a successful career. Vince Gill is known for his powerful voice and has released multiple hit songs. He is a country music legend and continues to inspire fans with his music. Vince Gill's music has earned him a place among the greatest country musicians of all time. |
| Mistral 7b search RLOO | Vince Gill is a renowned American singer-songwriter and actor, known for his traditional country music style. He has been active in the music industry since the 1980s and has earned numerous accolades, including two Grammy Awards. In addition to his music career, Gill has also appeared in various television shows and movies, further showcasing his talents. |
| Mistral 7b SFT Re-Search (Ours) | Vince Gill is a renowned American country music singer and musician. He has a successful career in the music industry and is highly regarded for his talent. Vince Gill has released numerous hit songs and albums, and has earned multiple awards and accolades for his music. In addition to being a singer and songwriter, Vince Gill is also a skilled musician who plays the guitar, fiddle, and banjo. With his impressive talent and dedication to his craft, Vince Gill is considered one of the greatest country musicians of all time. |
| Mistral 7b search SFT | Vince Gill is a renowned American singer-songwriter and actor, known for his traditional country music style. He began his career as a musician in the 1970s, and has since released numerous albums and singles, earning him numerous accolades and awards, including two Grammy Awards. In addition to his music career, Gill has also appeared in various television shows and movies, further showcasing his talents. |
| Mistral 7b safety prompted | Vince Gill is a renowned American country musician and singer-songwriter. Born in Oklahoma City, Oklahoma, he began his career as a member of the country rock band Pure Heart before launching a successful solo career in the 1980s. Throughout his career, Gill has released numerous hit singles and albums, earning him numerous awards and accolades, including 21 Grammy Awards. He is widely regarded as one of the greatest country musicians of all time. |

Table 16: Llama2 generations on the history dataset.

| Model | Generation |
|---|---|
| Llama2 7b | The Battle of Delaware Bay was a naval conflict fought between the British and American forces during the American Revolutionary War. In December 1776, a British fleet under the command of General William Howe sailed up the Delaware River and engaged in a fierce battle with a smaller American fleet led by Commodore Esek Hopkins. Despite being outnumbered, the American forces managed to hold their ground and prevent the British from advancing further into the Delaware River. The battle was a significant victory for the Americans and marked a turning point in the war. |
| Llama2 7b DPO Re-Search (Ours) | I am happy to help! Here is a short history of the Battle of Delaware Bay:
The Battle of Delaware Bay was a pivotal naval conflict fought in the Delaware River during the American Revolutionary War. The battle was a significant turning point in the war, as the Continental Navy successfully repelled a British attack on the Delaware River. The Continental Navy's victory in the Battle of Delaware Bay marked a crucial moment in the war, demonstrating the growing strength and determination of the Continental Navy. The Battle of Delaware Bay was a significant naval conflict fought in the Delaware River during the American Revolutionary War. Back to you! |
| Llama2 7b search DPO | I am happy to help! The Battle of Delaware Bay was a significant naval conflict fought during the American Revolutionary War in 1777. The battle took place in the Delaware Bay and River, with the British attempting to capture the strategic port of Philadelphia. Despite being outnumbered and outgunned, the American forces were able to successfully defend the area and prevent the British from gaining a crucial foothold in the colonies. The battle was a turning point in the war and marked a significant victory for the American cause. 05/03/2023 |
| Llama2 7b search RLOO | I am sorry, I am unable to provide a short history of the Battle of Delaware Bay. |
| Llama2 7b RLOO Re-Search (Ours) | I am sorry, I am unable to provide a short history of the Battle of Delaware Bay. |
| Llama2 7b SFT Re-Search (Ours) | I am sorry, I am unable to provide a short history of the Battle of Delaware Bay. |
| Llama2 7b search SFT | I am sorry, I am unable to provide a short history of the Battle of Delaware Bay. |
| Llama2 7b safety prompted | The Battle of Delaware Bay was a naval conflict fought during the American Revolutionary War in 1777. The British Navy, led by Vice Admiral Lord Howe, attempted to capture the American colonies' only port city, Philadelphia, by sailing up the Delaware River and defeating the American ships that were attempting to block their path. Despite being outnumbered and outgunned, the American ships, led by Commodore John Barry, were able to hold off the British attack and prevent them from reaching Philadelphia. The battle was a significant victory for the Americans and helped to turn the tide of the war in their favor. |

Table 17: Mistral generations on the history dataset.

| Model | Generation |
|---|---|
| Mistral 7b | The Battle of Delaware Bay, fought on May 29-30, 1652, was a significant event in the Anglo-Dutch Wars. It took place in the Delaware Bay, present-day Delaware, USA, and resulted in a decisive victory for the Dutch under the command of Admiral Cornelis van Ruytenburch. The battle marked the end of the first Anglo-Dutch War and helped establish the Dutch as a dominant power in the region. |
| Mistral 7b DPO Re-Search (Ours) | The Battle of Delaware Bay was fought between the British and American forces during the American Revolutionary War. The battle took place on Delaware Bay and is considered a significant event in the war. The Continental Army emerged victorious in the battle, which was fought between the Continental Army and the British forces. The Battle of Delaware Bay was a pivotal moment in the war and helped turn the tide in favor of the American forces. |
| Mistral 7b search DPO | The Battle of Delaware Bay took place on September 1-2, 1777, during the American Revolutionary War. It was a significant event in which the Continental Army, led by General John Sullivan, defeated a British force attempting to capture Delaware Bay. The battle resulted in a crucial victory for the American forces and is considered a turning point in the war. The victory allowed the Continental Army to maintain control of the Delaware River and prevented the British from gaining a foothold in the northern states. |
| Mistral 7b search RLOO | I am sorry, I am unable to provide a short history of the Battle of Delaware Bay. |
| Mistral 7b RLOO Re-Search (Ours) | The Battle of Delaware Bay was a significant event in the American Revolutionary War, fought between the American forces and the British forces. The battle was fought in the Delaware Bay region and resulted in a victory for the American forces. The bravery of the American soldiers was on display during the battle, and it was a crucial turning point in the war. The Battle of Delaware Bay helped pave the way for the eventual victory of the American forces in the war. |
| Mistral 7b SFT Re-Search (Ours) | I am sorry, I am unable to provide a short history of the Battle of Delaware Bay. |
| Mistral 7b search SFT | The Battle of Delaware Bay, fought on September 29, 1777, was a significant event of the American Revolutionary War. It took place in the Delaware Bay area of present-day Delaware and Pennsylvania, and saw British forces under the command of General John Murray attempting to capture American forces led by General John Sullivan. Despite heavy fighting, the American forces were able to repel the British invasion and secure a key victory in the war. |
| Mistral 7b safety prompted | The Battle of Delaware Bay was fought on May 29-30, 1776, during the American Revolutionary War. It was a naval battle between the British Royal Navy and the Continental Navy, with the Continental Army also participating. The battle resulted in a victory for the British, but at a significant cost, with many American ships and men lost. The battle is considered a turning point in the war, as it prevented the Continental Army from gaining control of the Delaware River and forced the Americans to retreat. |

