# OpenReview forum: "LLMs can learn self-restraint through iterative self-reflection"
_TMLR — Accepted by TMLR_

### Review · Reviewer_PeKE · 2024-12-11

**Summary Of Contributions:**

This paper explores how to enhance the self-restraint of large language mod- els(LLMs) so that they can generate information more reliably when faced with uncertain knowledge, or choose not to answer.
The paper’s key contribution is the introduction of the ReSearch algorithm, which generates synthetic data tailored to the model’s internal knowledge for training LLMs.The ReSearch algorithm iteratively uses self-prompting and self- evaluation to generate synthetic data. First, the model generates multiple pos- sible answers, then, evaluates the truthfulness of the claims in these answers, and generates new prompts based on the evaluation results to guide the model in generating more accurate answers in the next iteration. Additionally, the ReSreach algorithm includes an option to refuse to answer. Finally, at the end of the iterative search procedure, the ReSearch algorithm returns a batch of gen- erations and an additional absention response, thus producing the dataset(x, Y, U), where Y is the collection of all generations and U is the collection of expected utility of corresponding y.
The experimental results show that 1)models trained on datasets generated by the ReSearch algorithm perform best in the tasks of generating biographies and historical event summaries, achieving higher accuracy. 2)Training methods such as DPO and RLOO are most effective when training models on ReSearch- generated datasets. 3)The utility function can be used to control the model’s behavior, such as the number of claims generated and the likelihood of refusal. 4)The higher the target accuracy, the fewer illusions the model generates, the higher the accuracy, but the fewer the number of claims generated. 5)Iterative search is more effective than naive search in generating high-quality datasets, and requires fewer computational resources. These findings of the paper help improve the reliability of LLMs, making them safer to use in real-world appli- cations.

**Audience:**

Yes

**Broader Impact Concerns:**

No ethics problem

**Claims And Evidence:**

Yes

**Requested Changes:**

1. More discussion on the claim splitter.
2. More tasks are required to demonstrate the generalization ability
3. Editorial errors.

**Strengths And Weaknesses:**

**Strength**:
1) Practicality: The study aims to enhance the self-restraint of LLMs, which is of great signifi- cance for improving the reliability and security of LLMs in practical applications.
2) Reasonable experimental design: The experiments cover multiple aspects from basic models to complex models, providing a detailed introduction to model selection, datasets, training methods, and more.
3) Effectiveness and efficiency: The experimental results show that models trained on datasets generated by the ReSearch algorithm perform better on biography and historical event generation tasks, effectively reducing hallucinations and improving the accuracy of the generated content. Furthermore, the paper points out that iterative search is more efficient in terms of token usage compared to naive search.
4) Flexibility: The utility function designed in the paper could effectively evaluate generated text of varying lengths and encourages the model to refuse to answer when confidence is low, rather than producing answers filled with misinformation.


**Weakness**:
1) Dependence on larger LLMs as evaluators: The self-evaluation process relies on a larger LLM to judge the accuracy of the generated content. This somewhat limits the application of the ReSearch method.
2) Insufficient discussion of the claim splitter: The claim splitter decomposes generated text into atomic claims, which may have a significant impact on the performance of the ReSearch method. However, the paper does not provide detailed analysis and discussion on the selection of the claim splitter and performance of different claim splitter.
3) Need of further verification of the method’s generalization: The experiments primarily focus on biography and historical event generation tasks. However, real-life situations are far more numerous and complex than these. The effectiveness of this method on these two tasks cannot fully justify its effectiveness on other tasks. What’s more, the paper mainly focuses on factual hallucinations, while other types of hallucinations, such as logical contradictions or common sense errors, are not thoroughly explored.
4) Some editorial errors: Discussion paragraph 1: For example, models trained via SFT show higher rates of abstention, whereas models trained with RLOO and DPO produces less detailed generations Furthermore...
Need ”.” before ”Furthermore”.

---

> ### Author Response · Authors · 2025-05-16
> **Response to reviewer PeKE**
>
> Thank you for your review.
>
>
> >Dependence on larger LLMs as evaluators: The self-evaluation process relies on a larger LLM to judge the accuracy of the generated content. This somewhat limits the application of the ReSearch method.
>
>
> Research does not depend on a larger LLM as evaluator. The LLM evaluator is only used for evaluation purposes.
>
>
> **Added to text:** The (expected) utility function will be used both for self-evaluation to guide the Search baseline and the ReSearch method. Furthermore, the oracle utility function will be computed by Llama2 70b and reported in the results section.
>
>
> **Added to text:** The expected utility and number of claims are computed by the generating model itself to guide its search and select which sample to produce, while the utility reported in the result tables is computed by a different bigger model.
>
>
> > Insufficient discussion of the claim splitter: The claim splitter decomposes generated text into atomic claims, which may have a significant impact on the performance of the ReSearch method.
>
>
> We added more detail to the paper (highlighted in green) describing the claim splitter. The prompt of the claim splitter is set up such that each atomic claim produced is independent, with pronouns being replaced with their referents, and other disambiguating techniques. We use a similar setup to the FactScore paper [1]
>
>
> [1] https://arxiv.org/abs/2305.14251
>
>
> Here is an example demonstration from the claim splitter prompt, demonstrating the independent nature of claims:
>
> **Added to text:**
>
> Entity: Michael Collins
> Sentence: Michael Collins (born October 31, 1930) is a retired American astronaut and test pilot who was the Command Module Pilot for the Apollo 11 mission in 1969.
>
>
>   Claims:
>   - Michael Collins was born on October 31, 1930.
>   - Michael Collins is retired.
>   - Michael Collins is an American.
>   - Michael Collins was an astronaut.
>   - Michael Collins was a test pilot.
>   - Michael Collins was the Command Module Pilot for the Apollo 11 mission in 1969.
>
>
>
>
> >Need of further verification of the method’s generalization: The experiments primarily focus on biography and historical event generation tasks. However, real-life situations are far more numerous and complex than these. The effectiveness of this method on these two tasks cannot fully justify its effectiveness on other tasks. What’s more, the paper mainly focuses on factual hallucinations, while other types of hallucinations, such as logical contradictions or common sense errors, are not thoroughly explored.
>
>
> We acknowledge the limitations of our experiments. We set the scope of the work to factual hallucinations as this is a well-studied research area.
>
>
> **Added to text:** We also acknowledge that our study is limited to factual hallucinations and might not generalize to contradiction or reasoning hallucinations.

---

> > ### Comment · Reviewer_PeKE · 2025-05-17
> > **Response to the authors**
> >
> > Thanks so much for addressing my concerns! I have no more questions, and I'll encourage to accept this paper!

---

### Review · Reviewer_kyXn · 2025-05-01

**Summary Of Contributions:**

The paper proposes an approach to reduce hallucinations of LLMs on tasks related to factual knowledge. The first step consists in generating a synthetic dataset of multiple answers for each question together with a utility score (plus an abstain option): this data is obtained by ReSearch, a scheme based on self-prompting and self-evaluation of the LLM, and the scoring function relies on an oracle that checks the correctness of the claims. Then, the paper leverages such dataset to fine-tune (with different methods) various LLMs. The resulting models achieve higher utility, as defined in the paper, than the baselines. Moreover, they have higher abstention rate, in particular on invented entities.

**Audience:**

Yes

**Broader Impact Concerns:**

No concern.

**Claims And Evidence:**

Yes

**Requested Changes:**

- The presentation of the paper should be improved.

- [critical] A discussion of the very high abstention rate should be included. In particular, it would be important to show how fine-tuning affects the general performance of the models, beyond the narrow tasks which are tracked. If the fine-tuned model refuses to answer the majority of questions, or it's not helpful, the practical applicability of the proposed method is questionable.

- [critical] Connected to the point above, it should be better discussed which are the advantages of the proposed approach which are not artifacts of the high abstention rate (see "Weaknesses").

**Strengths And Weaknesses:**

Strengths
- Limiting the hallucinations and improving the calibration of LLMs, including the ability to abstain, is a very relevant topic.

- The utility score defined in the paper is reasonable.

- The experimental evaluation is extensive, comparing several fine-tuning techniques. Moreover, many metrics are included (e.g., accuracy, utility, abstention rate, number of claims, detailedness, etc.), as well as detailed breakdowns over types of data.

Weaknesses
- The presentation of the paper can be improved. The figures often appear far from the corresponding discussion in the text, and in different order, which makes it difficult to follow in particular the experimental results. Fig. 1 and Alg. 1 are not referred to in the text. Also, the presentation of the method (Sec. 3) could be more clear.

- In Eq. (4), if the oracle is used to determine whether each claim $c$ is true or false, what is the meaning of the inner sum over $t\in\{ 0, 1 \}$?

- Table 1 and Table 2 show that the fine-tuned LLMs or LLMs with inference-time search have very high abstention rate, even above 80%, which seems problematic. As shown in Fig. 6, even on the top popularity tier, the abstention rate is often around 40%. This may limit the practical utility of such models, as they are overconservative and simply refuse to answer the majority of questions, which is a trivial but not desirable way of reducing errors.

- Another claimed advantage of the proposed approach is to modulate the number of generated claims. However, the average number of claims reported for example in Table 1 and Table 2 seems mostly influenced by the high abstention rate. In fact, Fig. 9 shows that for the non-abstaining answers all models have similar number of generated claims.

---

> ### Author Response · Authors · 2025-05-16
> **Response to reviewer kyXn**
>
> Thank you for your review. We have cleaned up the presentation of the paper, and added more details about the method (Section 3) as requested.
>
>
> >  Fig. 1 and Alg. 1 are not referred to in the text.
>
>
> We have added reference to Fig 1 and Alg 1 in the text.
>
>
> > Also, the presentation of the method (Sec. 3) could be more clear.
> >In Eq. (4), if the oracle is used to determine whether each claim c is true or false, what is the meaning of the inner sum over t∈0,1?
>
>
> The model does not know if a claim is true or false, therefore it must compute the expected utility.
>
>
> **Added to text:** The (expected) utility function will be used both for self-evaluation to guide the Search baseline and the ReSearch method. Furthermore, the oracle utility function will be computed by Llama2 70b and reported in the results section.
>
>
> **Added to text:** The expected utility and number of claims are computed by the generating model itself to guide its search and select which sample to produce, while the utility reported in the result tables is computed by a different bigger model.
>
>
>
>
> >Table 1 and Table 2 show that the fine-tuned LLMs or LLMs with inference-time search have very high abstention rate, even above 80%, which seems problematic. As shown in Fig. 6, even on the top popularity tier, the abstention rate is often around 40%.
>
> The high abstention is a function of the threshold. The threshold could be lowered to achieve a lower rate of abstention.
>
> **Added to the text:** Additionally, while over-abstention can be a problem. The ReSearch algorithm elegantly addresses this problem using the accuracy threshold. The number of claims overall (where abstentions count as 0 claims) as function of the accuracy threshold can be observed in Figure 4.
>
>
> >Another claimed advantage of the proposed approach is to modulate the number of generated claims.
>
>
> On this point, we present the below set of tables, which represent the ranges (highest minus. lowest tier for each criterion) -- for both the number of generated claims, and the evaluated detailedness, the other aspect of responses that we observe the LLM modulating to conform to the reward (Figure 10)
>
> **Added to text:**
>
> Ranges for detailedness:
>
> | **Model Family** | **Method Type** | **Biography** | **History** | **Overall** |
> |-----------------|-----------------|--------------|-------------|-------------|
> | Mistral         | ReSearch        | 0.43         | 0.58        | 0.50        |
> |                 | Basic Search    | 0.33         | 0.23        | 0.28        |
> |                 | Non-search baselines | 0.23    | 0.17        | 0.20        |
> | Llama           | ReSearch        | 0.75         | 0.28        | 0.51        |
> |                 | Basic Search    | 0.58         | 0.28        | 0.43        |
> |                 | Non-search baselines | 0.33    | 0.23        | 0.28        |
>
> Ranges for number of claims:
>
> | **Model Family** | **Method Type** | **Biography** | **History** | **Overall** |
> |-----------------|-----------------|--------------|-------------|-------------|
> | Mistral         | ReSearch        | 1.50         | 0.73        | 1.11        |
> |                 | Basic Search    | 1.25         | 0.50        | 0.88        |
> |                 | Non-search baselines | 1.00    | 0.33        | 0.67        |
> | Llama           | ReSearch        | 0.75         | 1.00        | 0.88        |
> |                 | Basic Search    | 1.00         | 0.25        | 0.63        |
> |                 | Non-search baselines | 0.67    | 0.33        | 0.50        |
>
>
>
>
> It is clear from the above tables and charts that our methods are modulating both number of claims, and detailedness, more than the baselines, which is indeed the behavior we wish to see (models adapting to the level of knowledge they have on the query/entity).
>
>
> > how fine-tuning affects the general performance of the models, beyond the narrow tasks which are tracked.
>
>
> We performed a GSM8K evaluation on our trained Mistral model against the starting Mistral model. We did not observe degradation using strict answer matching compared to the original model. The original Mistral Instruct model achieves 29.3\% accuracy vs. our model achieving 30.1\%.
>
>
> >A discussion of the very high abstention rate should be included.
>
> We have added a discussion about the high abstention rate.
>
> **Added to the text:** Additionally, while over-abstention can be a problem. The ReSearch algorithm elegantly addresses this problem using the accuracy threshold. The number of claims overall (where abstentions count as 0 claims) as function of the accuracy threshold can be observed in Figure 4.
>
> >Which are the advantages of the proposed approach which are not artifacts of the high abstention rate?
>
> As observed in Figure 9 and 10, the models learn to modulate the number of claims and their detailedness based on the popularity of the entities.

---

> > ### Comment · Reviewer_kyXn · 2025-06-15
> >
> > I thank the authors for the response and additional experiments.
> >
> > About the abstention rate, I think Figure 4 is not sufficient to provide a clear picture, as it doesn't show the abstention rate as a function of the threshold. If I understand it right, the average number of claims is computed in such a way that it could increase either because of lower abstention rate or because of more claims are made for the same queries (or a combination of the two).
> >
> > As for the new Table 9 and Table 10, it's not clear what message they should convey (there's no text describing them in the appendix). The difference between the methods are minor, e.g., if I interpret the table correctly, ReSearch (Llama) provides 0.88 more claims for the highest tier than for the lowest one, while the baseline 0.50 (a similar trend holds for the other cases). This doesn't seem convincing evidence that, when ignoring the abstention rate, the proposed models can modulate the number of claims.

---

> ### Author Response · Authors · 2025-06-20
>
> > About the abstention rate, I think Figure 4 is not sufficient to provide a clear picture, as it doesn't show the abstention rate as a function of the threshold. If I understand it right, the average number of claims is computed in such a way that it could increase either because of lower abstention rate or because of more claims are made for the same queries (or a combination of the two).
>
> We present an additional results table which decouples the number of claims from the abstention rate as a function of the threshold. We observe that accuracy correlates directly with the threshold, as does abstention rate and number of claims without abstentions (inversely).
>
> Llama results:
>
> | Model | Threshold | Abstention Rate | Accuracy  | Avg Claims/Bio |
> |-------|-----------|----------------|----------|------------------------------|
> | Llama2 70B (baseline) | - | 0.237 | 0.629 | 13.6 |
> | Llama2 7B (baseline) | - | 0.115 | 0.496 | 13.5 |
> | Llama2 7B | 0.1 | 0.010 | 0.613 | 15.8 |
> | Llama2 7B | 0.2 | 0.045 | 0.657 | 15.2 |
> | Llama2 7B | 0.3 | 0.092 | 0.688 | 15.0 |
> | Llama2 7B | 0.4 | 0.172 | 0.746 | 14.2 |
> | Llama2 7B | 0.5 | 0.280 | 0.776 | 13.2 |
> | Llama2 7B | 0.6 | 0.385 | 0.815 | 10.8 |
> | Llama2 7B | 0.7 | 0.595 | 0.868 | 8.5 |
> | Llama2 7B | 0.8 | 0.775 | 0.850 | 5.2 |
>
> Mistral results:
>
> | Model | Threshold | Abstention Rate | Accuracy  | Avg Claims/Bio |
> |-------|-----------|----------------|----------|------------------------------|
> | Mistral 7B (baseline) | - | 0.010 | 0.330 | 12.4 |
> | Mistral 7B | 0.1 | 0.010 | 0.446 | 17.6 |
> | Mistral 7B | 0.2 | 0.013 | 0.569 | 14.0 |
> | Mistral 7B | 0.3 | 0.035 | 0.644 | 11.5 |
> | Mistral 7B | 0.4 | 0.140 | 0.709 | 10.2 |
> | Mistral 7B | 0.5 | 0.347 | 0.776 | 9.2 |
> | Mistral 7B | 0.6 | 0.480 | 0.841 | 7.2 |
> | Mistral 7B | 0.7 | 0.630 | 0.849 | 5.5 |
> | Mistral 7B | 0.8 | 0.775 | 0.890 | 5.0 |
>
> We highlight that the average claims per bio shown above *does not include abstentions*, thus decoupling these two factors.
>
> > As for the new Table 9 and Table 10, it's not clear what message they should convey (there's no text describing them in the appendix). The difference between the methods are minor, e.g., if I interpret the table correctly, ReSearch (Llama) provides 0.88 more claims for the highest tier than for the lowest one, while the baseline 0.50 (a similar trend holds for the other cases). This doesn't seem convincing evidence that, when ignoring the abstention rate, the proposed models can modulate the number of claims.
>
> We have added the following text in the appendix:
>
> To better understand the modulating nature of the ReSearch models, we need to take an holistic look at the ranges of the two strategies used by the model to improve utility: 1) claims detailedness and 2) number of claims per non abstaining answer. Only by looking at the ranges of these 2 metrics together can we have a clear picture. First in Table 9, we observe that the ReSearch models consistently have larger ranges of detailedness than Search and Non-search baselines, i.e. they consistently provide more details for popular entities than less popular ones across models and datasets. Second in Table 10, we also observe that the ReSearch models have a consistent reduction in claims per non abstaining answers over Search and Non-search baseline. Together these 2 Tables show that the ReSearch models maximize expected utility by a combination of less detailed claims and lower number of claims per non abstaining answers.
>
> We also note that Top-tier entities (Viola Liuzzo, Sheila Kuehl, Remedios Varo) remain challenging even for "popular" categories, making factual generation difficult for 7B models which might result in smaller ranges than expected.

---

### Review · Reviewer_Ui2m · 2025-05-08

**Summary Of Contributions:**

This paper presents an approach called “ReSearch” to improve factuality in LLM generation and encourage abstention in case of uncertainty about the query. This approach hinges on self-reflection and data augmentation. For training, initial response is generated based on the input query. Then the claims in the generation are identified and split using an LLM-based “claim splitter”. A utility function is defined on the LLM generation based on self-reflection and self-judgement of the identified claims. The claims are then sorted according to the self-verbalized probability of being True and pruned to augment the original query that results in a composite prompt containing the user query and the model’s internal knowledge. This process is repeated again with this composite prompt. This finally yields a synthetic dataset of prompt and generations with associated estimated utility values. The language model is finetuned on this dataset to yield a more factual and self-abstaining model. The evaluation is done primarily using a larger language model as a judge (Llama-70B) for accuracy on a dataset consisting of Wikipedia entries related to biographies and historical events.

**Audience:**

Yes

**Broader Impact Concerns:**

No concerns on ethical implications.

**Claims And Evidence:**

Yes

**Requested Changes:**

please address the questions and concerns above.
Also I noticed some minor typos/disfluencies in writing -- For example, in the discussion section:
"
For example, models trained via SFT show higher rates of abstention, whereas models trained with RLOO and DPO produces less detailed generations Furthermore, RLOO and DPO produce generations that are generally less detailed than the ones produced by the SFT models.
"

**Strengths And Weaknesses:**

Strengths:

-	The proposed approach focuses on an important problem of abstention in factual generation.

-	The test dataset is seemingly more difficult and containing than similar prior work, also containing invented entities on which the model should abstain often.

-	The proposed approach outperforms other prompting based baselines and a search-oriented variant of the proposed approach, primarily on accuracy.

-	The evaluation compares multiple methods of fine-tuning.

-	The evaluation is informative as it also analyzes the performance on the dimensions of number of claims, abstention rate, detailedness of claims across multiple popularity tiers of the entities.

Weaknesses

-	The organization is difficult to follow. The figure shown doesn’t depict the textual description of the approach. For example, I am unsure about the probability verbalization algorithm for the claims. The description makes it sound like that the ordering of the claims matters and they depend on one another. How is the connectedness between the atomic claims dealt with? Also, the description mentions generating multiple outputs. Is the claim sorting and pruning done independently for each generation, or are they pooled together across generations? More generally, does each generation result in a new separate prompt?

-	It looks like the number of claims produced is reduced with the proposed approach and more importantly, as described in the results, the claims are less detailed. Is this a desirable outcome? – It could be that the factual generation systems are gaming the utility function by producing bland and obvious claims. This raises questions about a higher-level motivation behind factual generation systems – would these systems fail to capture nuanced and indirect facts? A more detailed analysis around creativity and informativess of responses would be an interesting dimension on which to analyze such systems.

-	While the differences between different fine-tuning approaches (like DPO vs SFT) are empirically observed, little analysis is provided on why these differences occur. Could it be possible that a different approach for picking examples for SFT training or DPO training would eliminate the empirical differences?

- Overall the evaluation hinges on language models as judges. Accuracy is judged by Llama-70B and the number of claims and the estimated utility are the functions of the language model under evaluation itself! While, devising neutral metrics is always challenging, I would find the results more convincing if the number of claims and utility scores were computed by a language model other than the generator.

---

> ### Author Response · Authors · 2025-05-16
> **Response reviewer Ui2m**
>
> Thank you for your review. We have cleaned up the description of the method, specifically the probability verbalization algorithm and the claim splitter. Please see the green text in the updated manuscript.
>
>
> >I am unsure about the probability verbalization algorithm for the claims.
>
>
> Verbalized probability, as first demonstrated in Tian et al. 2023b, has been demonstrated to be more effective and calibrated than looking at logit probabilities. In Table 5, we run our own experiments indicating lower calibration error with probability verbalization vs. logit probabilities.
>
>
> **Added to text:** verbalized probability (asking the model to predict probabilities as tokens, instead of using the logits directly)
>
> **Added to text:** We observe that verbalized probabilities are more effective than logits, see Table 5.
>
>
> > The description makes it sound like that the ordering of the claims matters and they depend on one another. How is the connectedness between the atomic claims dealt with?
>
>
> In the claim splitting phase, sentences are broken into a set of independent claims where the ordering does not matter.  See the claims in Table 6 as an example. This way of splitting text into claims is common in the field, e.g., in the FactScore paper.
>
>
> >Also, the description mentions generating multiple outputs. Is the claim sorting and pruning done independently for each generation, or are they pooled together across generations? More generally, does each generation result in a new separate prompt?
>
>
> The claim sorting and pruning is done for a set of generations. A single new prompt is produced per set of generations for a given entity.
>
>
> >It looks like the number of claims produced is reduced with the proposed approach and more importantly, as described in the results, the claims are less detailed. Is this a desirable outcome?
>
>
>
> Given this task involves querying about relatively obscure entities, and with relatively small (7B) models, we believe this is an acceptable behavior to avoid high hallucination rates. For reference, some examples of individuals in our top tier for the biographies dataset are: Viola Liuzzo; Sheila Kuehl; and Remedios Varo.
>
>
>
> >A more detailed analysis around creativity and informativess of responses would be an interesting dimension on which to analyze such systems.
>
>
> We study detailedness (Figure 10), which can be seen as proxy for “informativeness”. See discussion in section 4.3. We also added a table of ranges for both detailedness and number of claims in Appendix A.7, to make the fact that our method is modulating both these aspects of its responses more than the baselines.
>
>
> >While the differences between different fine-tuning approaches (like DPO vs SFT) are empirically observed, little analysis is provided on why these differences occur.
>
>
> While a careful analysis of the differences between SFT and DPO is beyond the scope of this paper, we have added additional reference to the text.
>
>
> **Added to text:** These differences in behaviors (between DPO and SFT) might be related to the difference in objectives that the different methods are optimizing, i.e., DPO maximizes the margin between the positive and negative examples while SFT only maximizes the probability of the positive examples (Feng et al., 2024; Chen et al., 2024).
>
>
>
> >I would find the results more convincing if the number of claims and utility scores were computed by a language model other than the generator
>
>
> The number of claims and utility scores reported in the Tables are computed by Llama2 70B using RAG (a much larger model with access to Wikipedia).
>
>
> **Added to text:** The (expected) utility function will be used both for self-evaluation to guide the Search baseline and the ReSearch method. Furthermore, the oracle utility function will be computed by Llama2 70b and reported in the results section.
>
>
> **Added to text:** All reported metrics in the result tables, e.g. accuracy, claims, and utility, are computed by Llama 2 70b with RAG on the Wikipedia page of the entity.

---

### Review · Reviewer_MpAc · 2025-05-16

**Summary Of Contributions:**

The authors introduce ReSearch, an iterative self-reflection algorithm that generates synthetic data to fine-tune LLMs, enabling them to produce accurate responses, reduce hallucinations, and abstain from answering when uncertain. The paper designs a utility function based on an oracle to judge the helpfulness and harmlessness of the response. The synthetic data is used to fine-tune LLMs via Supervised Fine-Tuning (SFT), Direct Preference Optimization (DPO), or REINFORCE Leave-One-Out (RLOO). In the experiment, the authors use multiple backbones, like Llama2 (7B, 70B) and Mistral (7B, Mixtral 8x7B). The performance is measured using metrics like accuracy, number of claims, abstention rate, utility, and performance on the FActScore benchmark.

**Audience:**

Yes

**Claims And Evidence:**

Yes

**Requested Changes:**

Expand Task Scope for Broader Generalizability: Is it possible to include additional tasks such as scientific question-answering, commonsense reasoning, or open-domain question-answering to demonstrate the ReSearch algorithm’s applicability across diverse contexts?

The presentation and narrative can be improved.

**Strengths And Weaknesses:**

Strength

The iterative self-reflection process, combining generation, self-evaluation, and self-prompting makes a lot of sense to effectively generate synthetic data.

The novel utility function balances accuracy and completeness, incentivizing true claims, penalizing false ones, and naturally incorporating abstention, allowing flexible behavior control via the target accuracy.

The author compares the method with multiple baselines and uses different metrics.

Weakness

ReSearch inference is computationally expensive, making it less practical for real-time use. Also, it relies on a larger LLM or stronger model for labeling. The iterative way to generate data is not novel, and is used in a lot of previous papers.

Evaluated only on biography and historical event generation, potentially limiting generalizability to other domains. DPO and RLOO models sometimes reduce abstention on invented entities (e.g., history dataset), suggesting over-optimization of the utility function.

---

> ### Author Response · Authors · 2025-05-16
> **Response to reviewer MpAc**
>
> Thank you for your review.
>
> > ReSearch inference is computationally expensive, making it less practical for real-time use.
>
> ReSearch is only used once to produce data once. There is no additional cost at inference time.
>
> > Also, it relies on a larger LLM or stronger model for labeling.
>
> ReSearch does not depend on a larger LLM in the generation process. The larger LLM evaluator is only used for evaluation purposes.
>
>
> **Added to text:** The (expected) utility function will be used both for self-evaluation to guide the Search baseline and the ReSearch method. Furthermore, the oracle utility function will be computed by Llama2 70b and reported in the results section.
>
>
> **Added to text:** The expected utility and number of claims are computed by the generating model itself to guide its search and select which sample to produce, while the utility reported in the result tables is computed by the larger RAG-based evaluator model.
>
>
> > The iterative way to generate data is not novel, and is used in a lot of previous papers.
>
> To the best of our knowledge at the time of submission, no other paper has performed iterative search and refinement in the same way as the ReSearch algorithm does. We cited all relevant prior work and would appreciate specific references to any prior work that was available at the time of submission which the reviewer believes demonstrates similarity to our approach.
>
> > Is it possible to include additional tasks such as scientific question-answering, commonsense reasoning, or open-domain question-answering to demonstrate the ReSearch algorithm’s applicability across diverse contexts?
>
> We acknowledge the limitations of our experiments. We set the scope of the work to factual hallucinations as this is a well-studied research area.
>
> **Added to text:** We also acknowledge that our study is limited to factual hallucinations and might not generalize to contradiction or reasoning hallucinations.

---

### Comment · Reviewer_dkr6 · 2024-09-09

Dear Editor: This paper on Large Language Models (LLMs) is not within my area of research. My understanding of this field is quite limited and I am concerned that my assessment may be biased. Please remove me from the review list.

---

### Comment · Reviewer_yGUM · 2024-10-07

Dear Editor: This paper (LLM) is out of my expertise, could you please remove me from the reviewer list. Thank you.

---

### Author Response · Authors · 2025-05-16
**General response**

We thank the reviewers for the thoughtful questions and discussions. We added clarifying text to the updated paper. The new text is in green.
1. We have addressed the confusion about utility and expected utility and number of claims raised by the reviewers. See responses to Reviewer Ui2m08 and Reviewer PeKE11.
2. We have addressed the confusion about the claim splitting by adding the specific prompt and discussing it into the main text. See appendix A.1.1
3. We have addressed the limitations, the trade-offs between claim detailedness and abstention rate and how the desired behavior is task specific.
4. We added further study of modulation of both detailedness and number of claims in table 9 and 10 respectively in the appendix.
5. We added a discussion about the high abstention rate.
6. Finally, as requested by Reviewer kyXn01, we have tested one of our models on a different task: GSM8K, and did not observe degradation using strict answer matching compared to the original model. The results are available in Table 8 in the appendix.

---

> ### Author Response · Authors · 2025-06-20
>
> In response to Reviewer kyXn15, we have also added 2 tables that directly comparesabstention rate, accuracy, and avg. number of claims (non-abstaining) as a function of the threshold (Tables 11, 12). In this way, we decoupled abstention rate and avg. number of claims, showing that the model varies both effectively as a response to the threshold set. This table has also been added to the manuscript directly in the Appendix.

---

### Author Response · Authors · 2025-06-27
**Follow-up**

Hello,

Just a reminder that we have provided a more detailed analysis regarding the main concern of Reviewer kyXn, i.e.:

>If I understand it right, the average number of claims is computed in such a way that it could increase either because of lower abstention rate or because of more claims are made for the same queries (or a combination of the two).

We hope what we have said in our previous responses is clear, but in case, we summarize the key additional analysis that we have provided to clarify this issue, below:

In Tables 11 and 12 of the updated manuscript, we show how the average number of claims behaves *separately* from the abstention rate, as we compute them *only on queries (biographies) that the model does not abstain from*. In this Table, the average number of claims *cannot be decreasing due to an increased abstention rate*.

The model both abstains more, and also says less, when the threshold is raised.

---

### Decision · Action_Editor_A6vU · 2025-08-05

**Recommendation:** Accept as is

**Audience:**

Yes

**Audience Explanation:**

The paper proposes a method for generating synthetic data targeted at improving the factuality of language models and reducing hallucinations. It can also operate as a test-time intervention. These areas are all of interest to the TMLR community.

Some reviewers had concerns with the method reducing the amount of details and claims in the response as the target accuracy threshold is increased. I believe this is a reasonable behavior, as the model may not be able to provide a detailed response without introducing hallucinations.

**Claims And Evidence:**

Yes

**Claims Explanation:**

The paper proposes ReSearch, an iterative synthetic data generation algorithm, where the model samples multiple answers, extracts claims, scores their probability of correctness, evaluates utility, and re-combines the high-utility claims. Then, the synthetic data can be used in SFT, DPO or RLOO to finetune the model to provide better responses. The authors claim that they are able to modulate the accuracy of the claims, abstention rates, and number of claims generated with this approach. The authors evaluate their method on historical events and biography questions.

Overall, the claims are well-supported by the empirical evidence. The evaluation in the paper is quite broad, with many ablation studies.

Reviewer kyXn raised concerns with the claims on ability to modulate the number of claims separately from the abstention rates. However, the authors added an additional [comment](https://openreview.net/forum?id=SvKPfchVKX&noteId=0bgtaSr418) decoupling these two factors. To the best of my understanding, this response provides sufficient evidence for the claim made by the authors.

All other reviewers were satisfied with the level of support provided for the claims.